# Adoptive immunotherapy with transient anti-CD4 treatment enhances anti-tumor response by increasing IL-18Rα$^{hi}$ CD8$^+$ T cells

Seon-Hee Kim[1,2], Eunjung Cho[1], Yu I. Kim[3], Chungyong Han[1,3 ✉], Beom K. Choi [iD] [4 ✉] & Byoung S. Kwon[5,6 ✉]

Adoptive T cell therapy (ACT) requires lymphodepletion preconditioning to eliminate immune-suppressive elements and enable efficient engraftment of adoptively transferred tumor-reactive T cells. As anti-CD4 monoclonal antibody depletes CD4$^+$ immune-suppressive cells, the combination of anti-CD4 treatment and ACT has synergistic potential in cancer therapy. Here, we demonstrate a post-ACT conditioning regimen that involves transient anti-CD4 treatment (CD4$^{post}$). Using murine melanoma, the combined effect of cyclophosphamide preconditioning (CTX$^{pre}$), CD4$^{post}$, and ex vivo primed tumor-reactive CD8$^+$ T-cell infusion is presented. CTX$^{pre}$/CD4$^{post}$ increases tumor suppression and host survival by accelerating the proliferation and differentiation of ex vivo primed CD8$^+$ T cells and endogenous CD8$^+$ T cells. Endogenous CD8$^+$ T cells enhance effector profile and tumor-reactivity, indicating skewing of the TCR repertoire. Notably, enrichment of polyfunctional IL-18Rα$^{hi}$ CD8$^+$ T cell subset is the key event in CTX$^{pre}$/CD4$^{post}$-induced tumor suppression. Mechanistically, the anti-tumor effect of IL-18Rα$^{hi}$ subset is mediated by IL-18 signaling and TCR–MHC I interaction. This study highlights the clinical relevance of CD4$^{post}$ in ACT and provides insights regarding the immunological nature of anti-CD4 treatment, which enhances anti-tumor response of CD8$^+$ T cells.

[1] Division of Tumor Immunology, Research Institute, National Cancer Center, Goyang, Republic of Korea. [2] Department of Biomedical Laboratory Science, Catholic Kwandong University, Gangneung, Republic of Korea. [3] Graduate School of Cancer Science and Policy, National Cancer Center, Goyang, Republic of Korea. [4] Biomedicine Production Branch, Research Institute, National Cancer Center, Goyang, Republic of Korea. [5] Eutilex Institute for Biomedical Research, Eutilex Co., Ltd, Seoul, Republic of Korea. [6] Department of Medicine, Tulane University Health Sciences Center, New Orleans, LA, USA. ✉email: chungyong.han27@gmail.com; 11380@ncc.re.kr; bskwon@eutilex.com

mmunotherapy is an effective treatment modality for cancer, which involves the administration of immunomodulatory agents and immune cells[1,2]. Anti-CD4 monoclonal antibody, a prominent immunomodulatory agent, elicits robust anti-tumor immunity in various cancers by increasing tumor-infiltrating lymphocytes (TILs)[3–5] and promoting CD8[+] T-cell reactivity against tumor cell-derived antigens[6,7].

Adoptive T-cell therapy (ACT) harnesses ex vivo-primed tumor-reactive T cells (ex-T cells) (e.g., sorted T cells, TILs, and gene-engineered T cells) to directly attack malignant cells[8–11]. ACT is generally accompanied by lymphodepletion pre-conditioning that eliminates consumptive cytokine sinks and immune-suppressive cells for efficient grafting of ex-T cells[12–14]. However, the effects of lymphodepletion are transient and the inhibitory elements re-emerge eventually. As the anti-CD4 antibody depletes CD4-expressing cytokine sinks[3,15], adequate treatment after preconditioning in ACT has the potential to induce synergistic anti-tumor responses by enhancing the activity of ex-T cells and endogenous CD8[+] T cells (en-T cells; a newly repopulating subset after lymphodepletion).

Here we demonstrate that the application of anti-CD4 as a post-conditioning regimen (CD4[post]; transient CD4 depletion after ex-T-cell transfer), in combination with cyclophosphamide preconditioning (CTX[pre]), considerably enhances ACT efficacy. We show that CTX[pre]/CD4[post] accelerates the proliferation and effector function of ex-T and en-T cells by enriching the IL-18Rα[hi] CD8[+] T-cell subset, which is the key for increasing anti-tumor responses. We further reveal the characteristics of the IL-18Rα[hi] CD8[+] T-cell population and its generative mechanism in the CTX[pre]/CD4[post] regimen.

## Results

**CTX[pre]/CD4[post] enhances ACT efficacy.** To evaluate the kinetics and extent of CTX[pre]-induced lymphodepletion, which is generally included in ACT regimens[16], we treated C57BL/6 mice with different doses of CTX. The severity and duration of lymphopenia in lymphoid tissues correlated with the CTX dose (Fig. 1a and Supplementary Fig. 1a, b). On the basis of this result, we designed a regimen that utilizes CTX[pre], ex-T cells, and CD4[post], a combination that synergistically increases ACT efficacy (Fig. 1b). Three days after the B16-F10 challenge, C57BL/6 mice sequentially received CTX, ex vivo-primed Pmel-1 cells (melanoma-specific T-cell receptor (TCR)-transgenic CD8[+] T cells), and interleukin-2 (IL-2). The Pmel-1 cells were activated and cultured ex vivo for 2 days before infusion to be used as adoptively transferred ex-T cells in our ACT model. A transient treatment (for 5 weeks) of anti-CD4 started 7 days after CTX[pre], when the induced lymphopenia began to recover (Fig. 1a and Supplementary Fig. 1a, b). CTX[pre]/CD4[post] significantly increased tumor suppression and the survival rate of ex-T-cell-transferred mice (Fig. 1c, d). The absence of ex-T or CD4[post] (CTX/anti-CD4 or CTX/ex-T group, respectively) diminished the efficacy of the treatment. Notably, although not observed in the other groups, the group treated by ACT in the presence of CTX[pre]/CD4[post] (CTX/ex-T/anti-CD4 group) developed vitiligo, a sign of CD8[+] T-cell-induced melanocyte destruction[6,7], indicating the increased response of CD8[+] T cells (Fig. 1e, f). Significantly, five of ten mice showed completely suppressed melanoma development, which was not observed in the other groups (Fig. 1f). These results show the strong therapeutic potential of the CTX[pre]/ACT/CD4[post] combination in melanoma treatment.

**CTX[pre]/CD4[post] accelerates differentiation and proliferation of CD8[+] T cells.** We investigated the effect of CTX[pre]/CD4[post] on the CD8[+] T-cell populations, as the increase in CD8[+] T-cell activity in mice that developed vitiligo (Fig. 1e, f) was strongly associated with enhanced anti-tumor response. Two different CD8[+] T-cell populations, namely ex-T cells (ex vivo-primed tumor-reactive cells) and en-T cells (polyclonal endogenous cells), were analyzed separately.

First, we checked the phenotypical characteristics of the cells from lymphoid tissues (Fig. 2a). CTX[pre]/CD4[post]-experienced ex-T and en-T cells showed higher frequency of 4-1BB- and CD69-expressing cells (Fig. 2b and Supplementary Fig. 2a, b), indicating increase in TCR-mediated activation. Ex-T cells did not show any significant difference in memory status between the two regimens (Fig. 2c and Supplementary Fig. 2c). Ex vivo priming induced most ex-T cells to differentiate into the CD44[+] CD62L[−] effector memory subset irrespective of the conditioning regimen. In contrast, en-T cells, which had relatively higher proportion of CD44[−] CD62L[+] naive subset (>25%), displayed accelerated differentiation under CTX[pre]/CD4[post] compared with the CTX[pre] condition. We also examined the expression level of representative inhibitory receptors, the indicators of repeated antigen stimulation of CD8[+] T cells[17,18]. As was observed in memory-related markers (Fig. 2c), the expression profile of ex-T cells in the two groups were similar (Fig. 2d and Supplementary Fig. 2d). In contrast, en-T cells significantly upregulated PD-1, TIGIT, and LAG3 levels under CTX[pre]/CD4[post] compared with the CTX[pre] condition, implying the increased activation signal in lymphoid tissues.

Next, we examined the proliferation and persistence of CD8[+] T cells in CTX[pre]- and CTX[pre]/CD4[post]-experienced mice. Ex vivo-primed Thy1.1[+] Pmel-1 cells (acting as ex-T cells) and unstimulated polyclonal CD45.1[+] CD8[+] T cells (acting as en-T cells) were labeled with different fluorescent dyes before transfer to tumor-bearing or non-tumor-bearing C57BL/6 mice (Fig. 2e). This enabled us to evaluate the proliferation of both populations in the presence or absence of melanoma antigens (Supplementary Fig. 3a). CD4[post] began 7 days after CTX[pre] and it successfully depleted the CD4[+] T-cell population in CTX[pre]/CD4[post] group (Supplementary Fig. 3b). CTX[pre]/CD4[post] increased both the Thy1.1[+] and CD45.1[+] cell populations in the inguinal lymph nodes irrespective of the presence of tumor (Fig. 2f). Notably, CTX[pre]/CD4[post]-experienced tumor-bearing mice showed expansion burst of Thy1.1[+] Pmel-1 cells (Fig. 2g; 180.9-fold [Pmel-1], 25.7-fold [CD45.1[+]]). The expansion rate was lower in the spleen than in inguinal lymph nodes (Supplementary Fig. 4a, b). Analysis of division status and phenotype revealed that the combination of tumor inoculation and CTX[pre]/CD4[post] caused most Thy1.1[+] Pmel-1 cells to rapidly divide and differentiate into the CD44[+] CD62L[−] effector memory phenotype (Fig. 2h–j and Supplementary Fig. 4c–g). These results indicate that ex vivo-primed melanoma-specific CD8[+] T cells have a proliferation advantage, particularly in CTX[pre]/CD4[post]-experienced antigen-abundant tumor-draining lymph nodes. Notably, the proportion of extensively differentiating CD45.1[+] cells (>7 division, CD44[+] CD62L[−]) increased in the lymph node and spleen, irrespective of tumor presence (Fig. 2h–j and Supplementary Fig. 4c–j).

**CTX[pre]/CD4[post] increases effector function and tumor reactivity of endogenous CD8[+] T cells.** Previous studies have demonstrated that depletion of CD4[+] cells is associated with increase in effector function, tumor reactivity, and intratumoral infiltration of CD8[+] T cells[3–5,7]. To determine whether CTX[pre]/CD4[post] yielded the same results, we analyzed these functional characteristics in ex-T and en-T cells (Fig. 3a). Effector function was evaluated in terms of the cell proportion that simultaneously secretes multiple effector cytokines (interferon-γ, tumor necrosis factor, and IL-2; i.e., polyfunctional). Similar to the results of memory status analysis (Fig. 2c), ex-T cells did not differ significantly between the CTX[pre] and CTX[pre]/CD4[post] groups (Fig. 3b and Supplementary Fig. 5a, b).

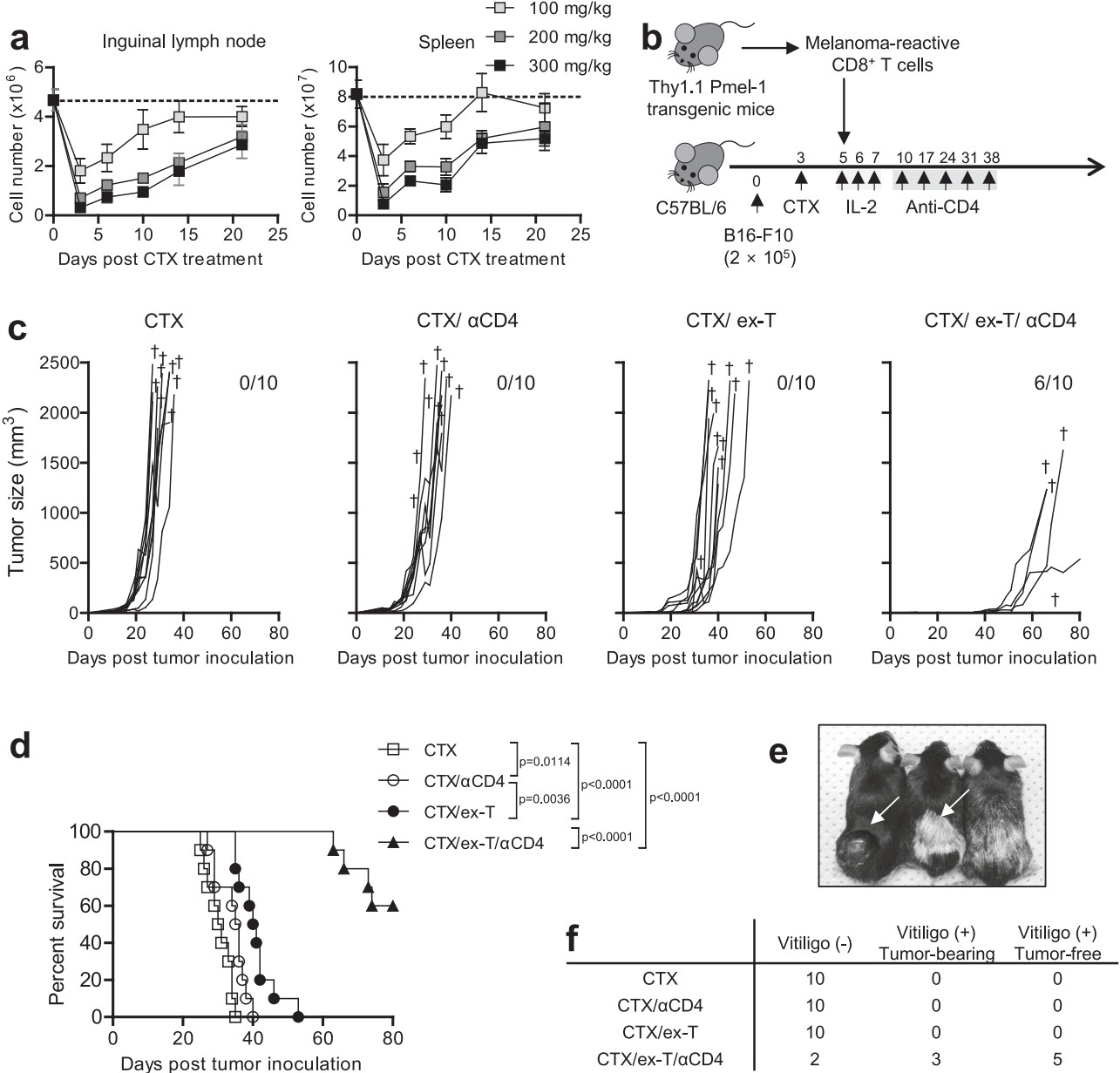

**Fig. 1 Anti-CD4 post conditioning increases anti-melanoma responses in adoptive T-cell therapy.** Efficacy evaluation of adoptive T-cell therapy in C57BL/6 mice. **a** Cell count from inguinal lymph node (left) and spleen (right) of cyclophosphamide (CTX)-treated C57BL/6 mice. Mice were intraperitoneally injected with the indicated doses (mg/kg) of CTX. $n = 3$ mice per each point. Error bars indicate means ± SD. **b** Schematic of the experiment. Injected cell number: B16-F10, $2 × 10^5$/mouse; treated ex vivo-primed Thy1.1$^+$ Pmel-1 CD8$^+$ T (ex-T) cells, $2 × 10^6$/mouse. Two days prior to ex-T-cell transfer, mice were pre-conditioned with 300 mg/kg CTX. Mice with anti-CD4 post conditioning were treated every week with anti-CD4 antibody from day 10 for 5 weeks. **c** Tumor growth curves are shown. Each curve indicates longitudinal changes of an individual mouse. † indicates the death of an individual mouse at the indicated time point. The number of mice surviving up to day 80 is shown. **d** Survival rate per group is indicated; two-tailed log-rank (Mantel–Cox) tests were used to determine statistical significance. **e** Representative images of mice with and without vitiligo. **f** Number of mice with vitiligo and having no sign of tumor on day 80. **c–f** $n = 10$ mice/group. αCD4, anti-CD4 antibody. Source data are provided as a Source Data file.

Highly differentiated ex-T cells showed stronger polyfunctional response than en-T cells irrespective of the presence of CD4$^{post}$. In contrast, the ratio of polyfunctional en-T cells was significantly higher in CTX$^{pre}$/CD4$^{post}$ than in the CTX$^{pre}$ group (cell frequency of two or more cytokine expression; CTX$^{pre}$ = 15.88 ± 1.74%, CTX$^{pre}$/CD4$^{post}$ = 55.46 ± 9.16%; mean ± SD; $p < 0.0001$; two-tailed unpaired Student's $t$-test).

Unlike ex-T cells, the population size of tumor-reactive en-T cells is limited at the time of tumor inoculation. To evaluate the change in tumor reactivity after CTX$^{pre}$/CD4$^{post}$-induced

differentiation, we checked CD25 expression and the killing activity of en-T cells after exposure to B16-F10. Ex-T, which is highly tumor reactive, upregulated CD25 expression after B16-F10 exposure in both CTX$^{pre}$ and CTX$^{pre}$/CD4$^{post}$ groups (Fig. 3c and Supplementary Fig. 2e). Notably, en-T cells showed a significantly higher ratio of CD25-expressing cells in CTX$^{pre}$/CD4$^{post}$ than in the CTX$^{pre}$ group, implying that anti-CD4 treatment heightened the tumor reactivity of CD8$^+$ T cells. This was further corroborated by the result of the in vitro killing assay, in which the B16-F10 killing efficiency of the CTX$^{pre}$/CD4$^{post}$-

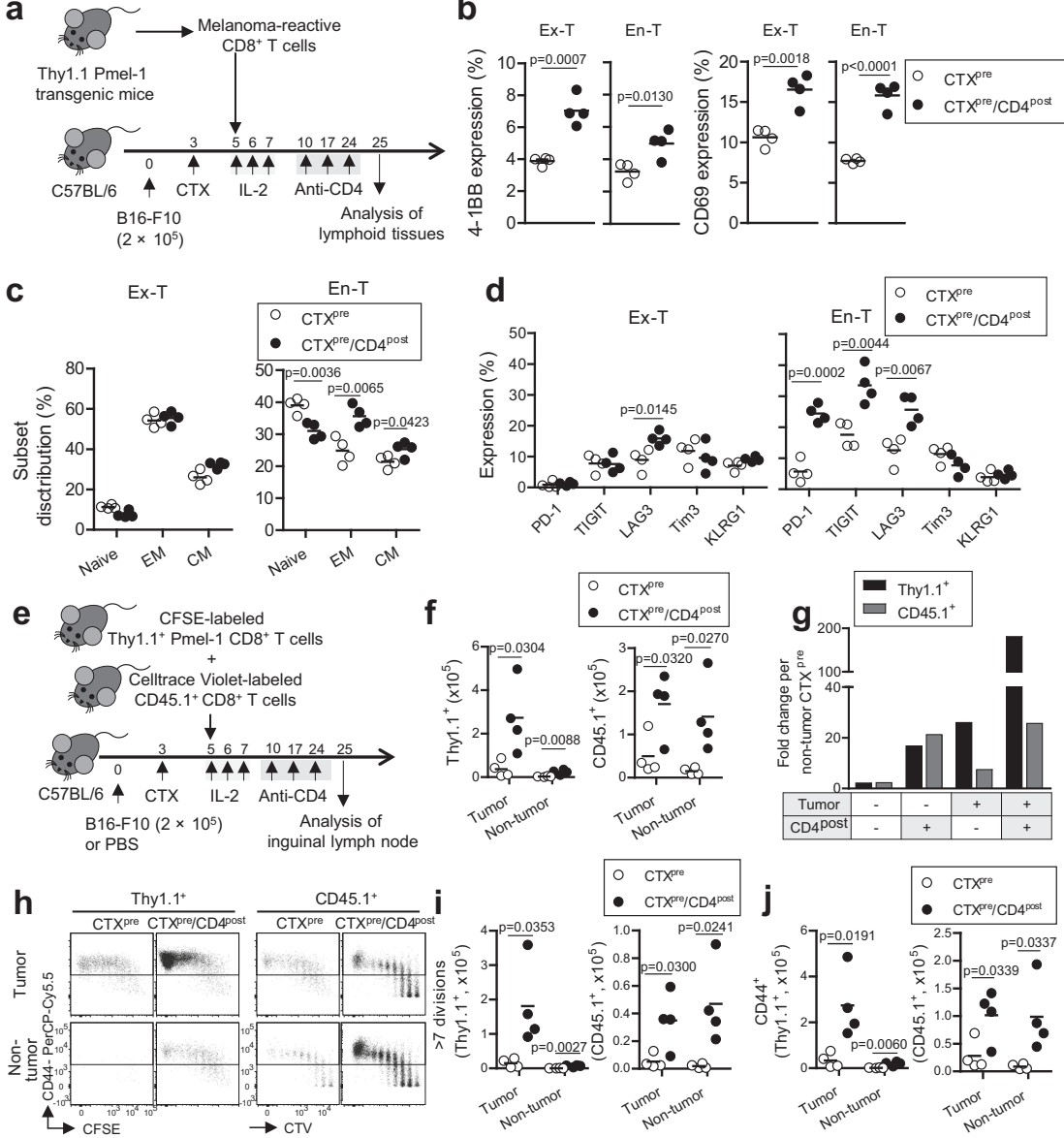

**Fig. 2 Anti-CD4 post conditioning accelerates proliferation and differentiation of ex vivo-primed and endogenous CD8$^+$ T cells. a** Schematic of the experiment. Injected cell number: B16-F10, $2 \times 10^5$/mouse; ex vivo-primed Thy1.1$^+$ Pmel-1 CD8$^+$ T (ex-T) cells, $2 \times 10^6$/mouse. **b–d** Marker expression and subset analysis of ex-T and endogenous CD8$^+$ T (en-T) cells. **c** The proportion of naive (CD62L$^+$ CD44$^-$), EM (effector memory; CD62L$^-$ CD44$^+$), and CM (central memory; CD62L$^+$ CD44$^+$) subsets were calculated. **e–j** Analysis of the proliferation of ex vivo-primed Thy1.1 Pmel-1 CD8$^+$ T (Thy1.1$^+$) cells and unstimulated polyclonal CD8$^+$ T (CD45.1$^+$) cells in the presence or absence of tumor antigen. **e** Schematic of the experiment. Injected cell number: B16-F10, $2 \times 10^5$/mouse; Thy1.1$^+$, $1 \times 10^6$/mouse; CD45.1$^+$, $1 \times 10^6$/mouse. **f** The cell count of Thy1.1$^+$ and CD45.1$^+$ cells. **g** Fold change in cell number over the average of the non-tumor CTX$^{pre}$ group. **h** Representative flow cytometry images of Thy1.1$^+$ and CD45.1$^+$ cells analyzed for CD44 expression. The number of cells with more than seven divisions (**i**) and of CD44 expression (**j**) were calculated. $n = 4$ mice per group. Each symbol indicates the calculated value of an individual mouse. Horizontal bars indicate means. Two-tailed unpaired Student's $t$-test. CD4$^{post}$, anti-CD4 post conditioning; CFSE, carboxyfluorescein succinimidyl ester; CTV, Celltrace Violet; CTX$^{pre}$, cyclophosphamide preconditioning; PBS, phosphate-buffered saline. Source data are provided as a Source Data file.

exposed en-T cells was higher than that of CTX$^{pre}$-exposed en-T cells (Fig. 3d). Biased T-cell activation alters the TCR repertoire, which promotes clonal expansion and enrichment of certain TCR clonotypes[19–21]. Therefore, we analyzed the Shannon evenness of the TCR repertoire, which denotes the extent of equal distribution of the clonotypes, with lower values indicating that some T-cell clones proliferated preferentially[21]. The CTX$^{pre}$/CD4$^{post}$ group had significantly lower evenness than the CTX$^{pre}$ group (Fig. 3e; $p = 0.03$ for TCR$\alpha$ and TCR$\beta$; Mann–Whitney $U$-test), indicating the enrichment of certain T-cell clonotypes.

Next, we analyzed the intratumoral CD8$^+$ T cells in mice that were inoculated with high dose of B16-F10 to establish measurable tumor volume in both CTX$^{pre}$ and CTX$^{pre}$/CD4$^{post}$ groups (Fig. 3f). As expected from the effect of CTX$^{pre}$/CD4$^{post}$ on T-cell expansion and tumor reactivity (Figs. 2f, g and 3c–e), CTX$^{pre}$/CD4$^{post}$ increased intratumoral infiltration of ex-T and en-T cells (Fig. 3g). Phenotypic analysis showed that CTX$^{pre}$/CD4$^{post}$-experienced en-T cells were more differentiated and included higher proportion of PD-1$^+$ cells than the CTX$^{pre}$-exposed group (Fig. 3h, i and Supplementary Fig. 2f, g). Highly tumor-reactive

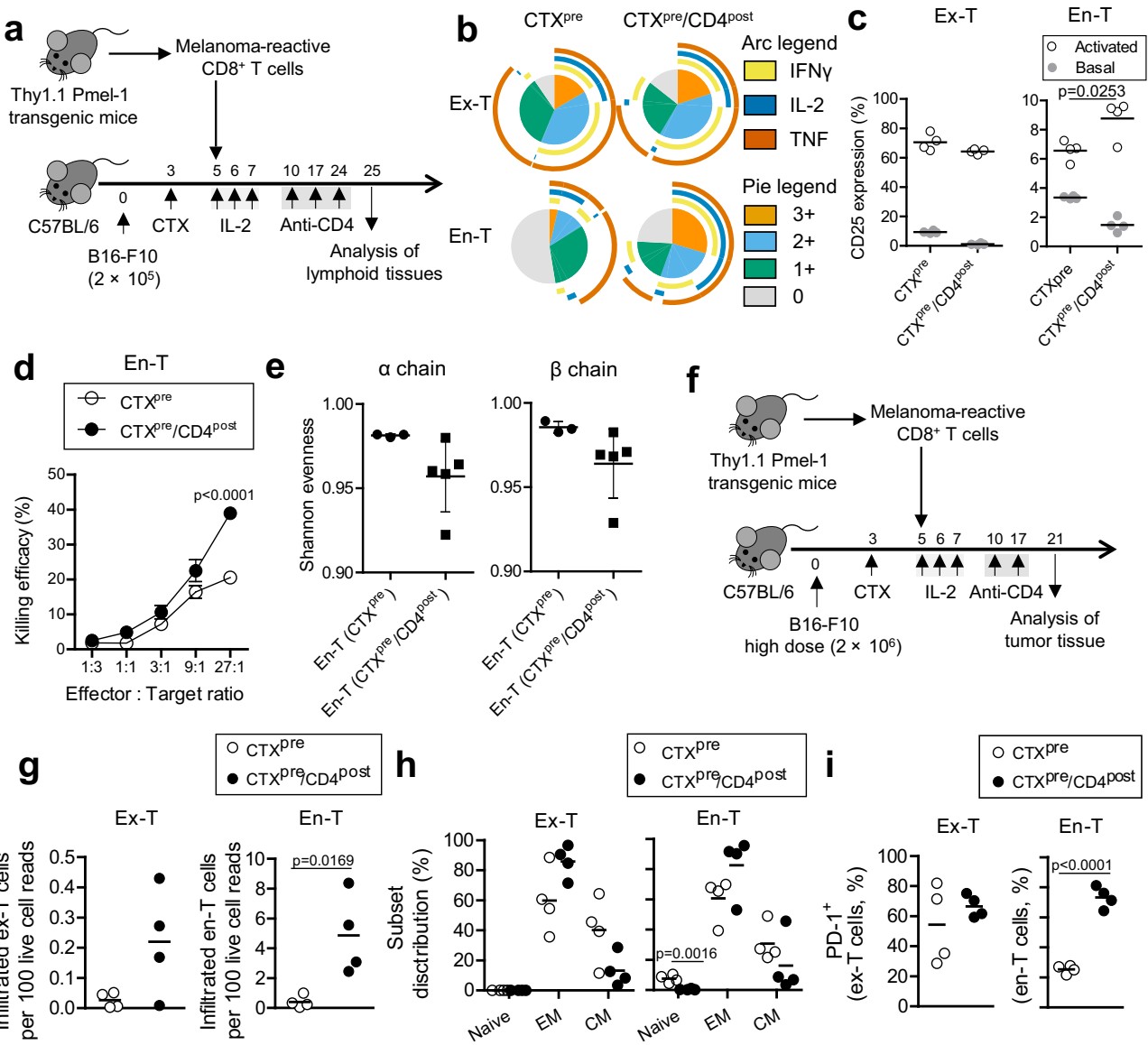

**Fig. 3 Anti-CD4 post conditioning increases tumor reactivity of endogenous CD8$^+$ T cells. a** Schematic of the experiment. Injected cell number: B16-F10, $2 \times 10^5$/mouse; ex vivo-primed Thy1.1$^+$ Pmel-1 CD8$^+$ T (ex-T) cells, $2 \times 10^6$/mouse. **b** Analysis of polyfunctionality. Cells were stimulated and analyzed for cytokine expression as in Supplementary Fig. 5a, b. SPICE v6.0 was used to draw the pie chart. Arc length indicates the proportion of cells expressing the corresponding marker. Pie area indicates the proportion of cells expressing the given number of functional markers. $n = 5$ mice per group. **c** CD25 expression was analyzed in lymphocytes incubated with (activated) or without (basal) B16-F10 for 16 h. **d** The killing efficiency of endogenous CD8$^+$ (en-T) cells against B16-F10 was evaluated. Cells from four mice of each group were pooled and assayed in biological triplicates. Error bars indicate means ± SEM. Two-way ANOVA with Bonferroni post hoc test was used to determine statistical significance. **e** Shannon evenness (normalized Shannon–Wiener index) of TCR clonotype analysis. TCR α-chain (left) and TCR β-chain (right) of en-T cells were analyzed. CTX$^{pre}$, $n = 3$ mice; CTX$^{pre}$/CD4$^{post}$, $n = 5$ mice. Error bars indicate means ± SD. **d**, **e** En-T cells isolated as in Supplementary Fig. 3e, f were used. **f–i** Characterization of tumor-infiltrating CD8$^+$ T cells. **f** Schematic of the tumor tissue experiments. Injected cell number: B16-F10, $2 \times 10^6$/mouse; ex-T cells, $2 \times 10^6$/mouse. **g** The count of ex-T and en-T cells among 100 live cells from tumor tissues were calculated using flow cytometry data. **h** The proportion of naive (CD62L$^+$ CD44$^-$), EM (effector memory; CD62L$^-$ CD44$^+$), and CM (central memory; CD62L$^+$ CD44$^+$) subsets were calculated. **i** The frequency of cells expressing PD-1 in ex-T and en-T cells are shown. **c**, **g–i** $n = 4$ mice per group. **c**, **e**, **g–i** Each symbol indicates the calculated value of an individual mouse. Horizontal bars indicate means. **c**, **g–i** Two-tailed unpaired Student's $t$-test. CD4$^{post}$, anti-CD4 post conditioning; CTX$^{pre}$, cyclophosphamide preconditioning. Source data are provided as a Source Data file.

ex-T cells showed 54.3 ± 26.3% and 66.5 ± 7.4% (mean ± SD) PD-1 positivity in CTX$^{pre}$ and CTX$^{pre}$/CD4$^{post}$ groups, respectively.

Collectively, these results suggest that the use of anti-CD4 treatment increases effector function, tumor reactivity, and intratumoral infiltration of endogenous CD8$^+$ T cells in this ACT combination model.

**CTX$^{pre}$/CD4$^{post}$ enriches IL-18Rα$^{hi}$ CD8$^+$ T cells.** So far, the identified factors that potentially affected ACT efficacy following CTX$^{pre}$/CD4$^{post}$ involved expansion of ex-T cells and increased tumor reactivity of en-T cells. To further define the factors that contribute to anti-tumor response in the current regimen, we investigated the gene expression profile of CD8$^+$ T cells that were

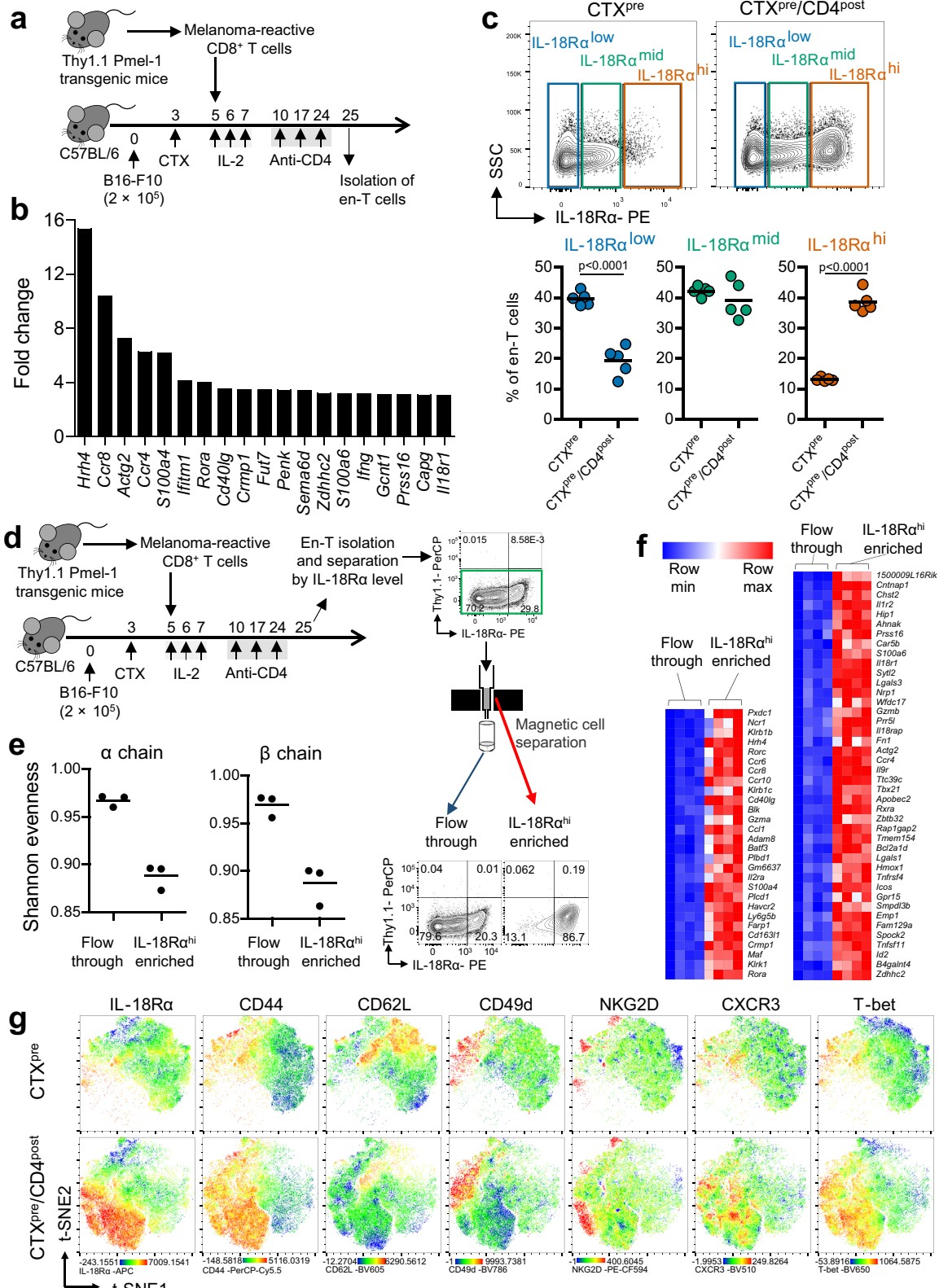

exposed to the milieu immunomodulated by CTX^pre/CD4^post. We used en-T cells for the analysis (Fig. 4a), as they exhibited more severe phenotypic/functional changes than ex-T cells. Among the genes upregulated over twofold in the CTX^pre/CD4^post group compared with that in the CTX^pre group (Supplementary Data 1), 21% were associated with T-cell activation, differentiation, and migration (20 of 95 genes; *Ccl1, Ccr4, Ccr6, Ccr8, Cd40lg, Epas1,*

*Fut7, Gcnt1, Gzmb, Icos, Ifitm1, Ifng, Il18r1, Lgals1, Penk, Rora, Rorc, S100a4, Sema6d,* and *Tigit*), consistent with the results of phenotypic analysis (Fig. 2c, d, h–j and Supplementary Fig. 4c–j). Importantly, the CTX^pre/CD4^post-experienced en-T cells exhibited more than threefold upregulation of *Ifng* and *Il18r1* (Fig. 4b), which encode key components in IL-18–IL-18R signaling[22,23]. Protein expression analysis revealed that among the three

**Fig. 4 Anti-CD4 post conditioning enriches IL-18Rα$^{hi}$ endogenous CD8$^+$ T cells. a** Schematic of the experiment. Injected cell number: B16-F10, $2 \times 10^5$/mouse; ex vivo-primed Thy1.1$^+$ Pmel-1 CD8$^+$ T (ex-T) cells, $2 \times 10^6$/mouse. **b** Differentially expressed genes in microarray analysis. Genes that were upregulated (>3-fold) in CTX$^{pre}$/CD4$^{post}$ compared with that in CTX$^{pre}$-experienced endogenous CD8$^+$ T (en-T) cells are indicated. En-T cells isolated as in Supplementary Fig. 3f were used. For each group, en-T cells from five mice were pooled and used for analysis. **c** IL-18Rα expression in en-T cells was analyzed. $n = 5$ mice per group. Two-tailed unpaired Student's $t$-test. **d–f** Analysis of TCR distribution and gene expression profiles of IL-18Rα$^{hi}$ en-T cells. **d** Schematic of analyte preparation. Injected cell number: B16-F10, $2 \times 10^5$/mouse; ex-T cells, $2 \times 10^6$/mouse. On day 25, IL-18Rα$^{hi}$-enriched en-T cells were isolated via magnetic cell separation. **e** Shannon evenness (normalized Shannon–Wiener index) of TCR α-chain (left) and β-chain (right) are shown. $n = 3$ mice per group. **f** Whole transcriptome analysis. Genes that were upregulated (>3-fold) in IL-18Rα$^{hi}$-enriched en-T cells compared with en-T cells in the flow-through are indicated. $n = 4$ mice per group. **g** Multi-color flow cytometry analysis of en-T cells. Analytes prepared as in **a** were stained with the indicated markers and analyzed. Dimensionality reduction (t-SNE; $t$-distributed stochastic neighbor embedding) was performed for visualization. **c, e** Each symbol indicates an individual mouse. Horizontal bars indicate means. xCD4$^{post}$, anti-CD4 post conditioning; CTX$^{pre}$, cyclophosphamide preconditioning. Source data are provided as a Source Data file.

populations that were grouped on the basis of the level of IL-18Rα expression, IL-18Rα$^{hi}$ en-T cells were significantly enriched in the mice exposed to CTX$^{pre}$/CD4$^{post}$ (Fig. 4c).

Next, we analyzed the subset-specific characteristics by comparing cells that were magnetically enriched for high IL-18Rα-expression with other cells in the flow-through (Fig. 4d). Although the two groups were sampled from the same CTX$^{pre}$/CD4$^{post}$-exposed CD8$^+$ T cells, TCR repertoire analysis showed that Shannon evenness was lower in the IL-18Rα$^{hi}$ group (Fig. 4e). Further gene expression analysis defined the molecular profile of the IL-18Rα$^{hi}$ en-T-cell population (Supplementary Data 2). We found that the genes associated with T-cell activation/differentiation, of which some were upregulated in CTX$^{pre}$/CD4$^{post}$-exposed CD8$^+$ T cells (Fig. 4b), were upregulated more than three times in IL-18Rα$^{hi}$-enriched cells (Fig. 4f). In addition, the upregulated genes included those encoding innate-like T-cell-related markers (*Pxdc1*, *Hrh4*, *Rorc*, *Ccr6*, *Cd40lg*, and *Blk*) and natural killer cell receptors (*Ncr1*, *Klrb1b*, *Klrb1c*, and *Klrk1*), which not only increased after typical TCR-dependent stimulation, but were also upregulated via TCR-independent/cytokine-mediated T-cell activation, such as bystander activation[24–26]. Multidimensional protein expression analysis defined the IL-18Rα$^{hi}$-associated phenotypic feature, which is characterized by CD44$^+$ CD62L$^-$ CD49d$^-$ CXCR3$^+$ T-bet$^+$ (Fig. 4g).

**Enhanced anti-tumor activity of IL-18Rα$^{hi}$ CD8$^+$ T cells is mediated by TCR/IL-18 signaling.** IL-18 signaling has been reported as an important mediator of T-cell effector function in cancer[27]. In addition, the high prevalence of IL-18Rα$^+$ cells in tumor was associated with better prognosis[28]. We investigated the anti-tumor potential of IL-18Rα$^{hi}$ CD8$^+$ T cells that were enriched in mice exposed to CTX$^{pre}$/CD4$^{post}$. The CTX$^{pre}$/CD4$^{post}$-exposed en-T cells were categorized into IL-18Rα$^{low}$, IL-18Rα$^{mid}$, and IL-18Rα$^{hi}$ subsets, and their polyfunctionality was assessed. Among the three groups, the IL-18Rα$^{hi}$ cells exhibited the highest polyfunctionality (Fig. 5a and Supplementary Fig. 5c, d), indicating that their anti-tumor potential was higher than those of the other two subsets.

So far, IL-18Rα$^{hi}$ en-T cells have shown skewed TCR repertoire (Fig. 4e), expressed high level of natural killer cell receptors (Fig. 4f), and were polyfunctional (Fig. 5a). These observations raised two questions: "Does the IL-18Rα$^{hi}$ subset show strong anti-tumor activity in vivo?" and "Is the interaction between major histocompatibility complex class I (MHC I) and TCR dispensable for anti-tumor effect?". Hence, we evaluated the in vivo anti-tumor activity of the IL-18Rα$^{hi}$ subset that was isolated from CTX$^{pre}$/CD4$^{post}$-experienced mice. To assess the subset-related effect of IL-18Rα$^{hi}$ population, en-T cells were enriched based on IL-18Rα expression and transferred into B16-F10-bearing *Rag1* knockout (KO) mice (Fig. 5b and Supplementary Fig. 3h). In the long term, IL-18Rα$^{hi}$ en-T cells were not

beneficial for survival rate and tumor progression (Supplementary Fig. 6a, b). Instead, we observed weak short-term effect (~27 days) of the subset in terms of tumor suppression (Fig. 5c and Supplementary Fig. 6c). Notably, treatment with an anti-MHC I antibody abrogated the suppressive effect, implying that this event is mediated by the interaction between MHC I and TCRs, which recognize antigens expressed on B16-F10 melanoma.

CTX$^{pre}$/CD4$^{post}$ treatment created a milieu in which en-T cells were prone to differentiate into a polyfunctional IL-18Rα$^{hi}$ subset; the subset elicited short-term effector function via MHC I–TCR interaction. We hypothesized that ex-T cells undergo the same differentiation that accompanies IL-18Rα expression in the CTX$^{pre}$/CD4$^{post}$-induced milieu. Intriguingly, CTX$^{pre}$/CD4$^{post}$ treatment enriched the similarly polyfunctional IL-18Rα$^{hi}$ CD8$^+$ T cells in the ex-T population (Fig. 5d, e and Supplementary Figs. 2h and 5e, f). In response to the specific gp100$_{25–33}$ peptide, IL-18Rα$^{hi}$ ex-T cells secreted higher level of effector cytokines than the IL-18Rα$^{low/mid}$ ex-T cells (Fig. 5f). Notably, the response was augmented by the addition of IL-18, suggesting the crucial role of IL-18 signaling in the anti-tumor activity of this subset. To evaluate tumor-suppressive activity in vivo, we isolated IL-18Rα$^{hi}$ ex-T cells from CTX$^{pre}$/CD4$^{post}$-experienced mice and transferred them to B16-F10-bearing C57BL/6 mice (Fig. 5g). Furthermore, we also administered the sorted cells to *Il18r1* KO mice, to evaluate the effect of increased accessibility to IL-18 on the transferred IL-18Rα$^{hi}$ or IL-18Rα$^{low/mid}$ ex-T cells. As was observed in en-T cells (Fig. 5c and Supplementary Fig. 6a–c), IL-18Rα$^{hi}$ ex-T cells showed only short-term effect (~32 days) in this model (Fig. 5h and Supplementary Fig. 7a–c). The effect was reduced in mice treated with IL-18-blocking antibody, supporting the role of IL-18 observed in the in vitro experiment (Fig. 5f). IL-18Rα$^{low/mid}$ ex-T cells showed lower anti-tumor activity than the IL-18Rα$^{hi}$ population and did not differ significantly from the IL-18-blocked IL-18Rα$^{hi}$ ex-T group. Notably, compared with C57BL/6 wild-type mice, *Il18r1* KO recipient mice, which lack IL-18R-expressing en-T cells, showed larger difference in the effect between the transfer of IL-18Rα$^{low/mid}$ and IL-18Rα$^{hi}$ ex-T cells. However, in long-term analysis, significant differences were not observed in tumor suppression and survival rate (Supplementary Fig. 7a, b), implying that IL-18Rα$^{hi}$ ex-T cells themselves have only short-term benefit without the continuous generation of the subset.

CTX$^{pre}$/CD4$^{post}$ changed the functional profile of en-T/ex-T cells, increasing the proportion of the IL-18Rα$^{hi}$ subset (Figs. 4c and 5d). However, despite the high effector potential, direct transfer of the cell subset exhibited only short-term effect in tumor suppression (Fig. 5c, h and Supplementary Figs. 6 and 7). These results implied that the CTX$^{pre}$/CD4$^{post}$-induced milieu, which continuously generates IL-18Rα$^{hi}$ CD8$^+$ T cells, rather than the short-term presence of the cells, drove the strong anti-tumor response in CTX$^{pre}$/ACT/CD4$^{post}$-experienced mice (Fig. 1c, d). Therefore, we investigated the anti-tumor effect of

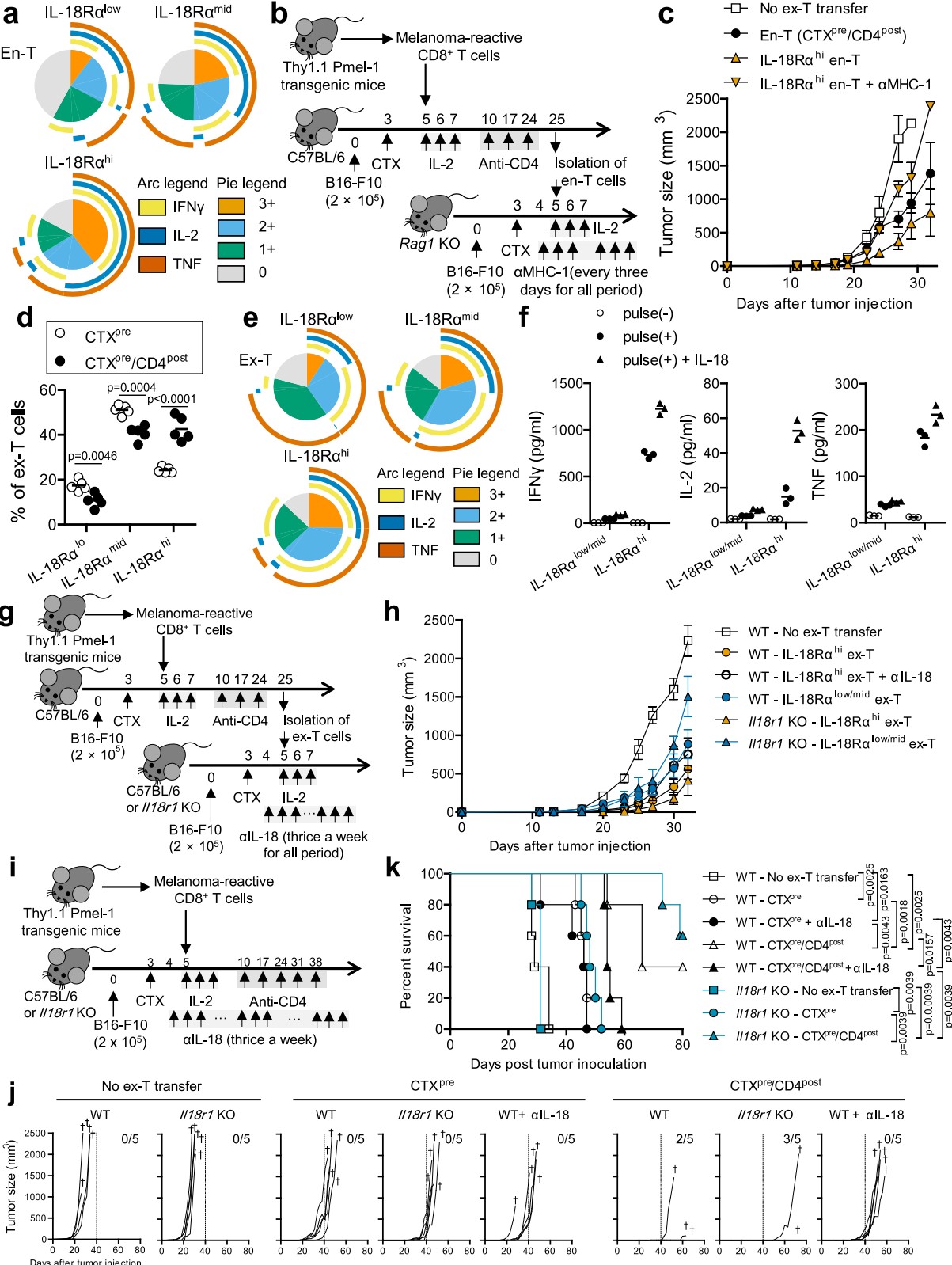

the CTX[pre]/ACT/CD4[post]-induced milieu in the long-term by blocking the IL-18 signal, which played a crucial role in the function of IL-18Rα[hi] CD8[+] T cells in vitro and in vivo (Fig. 5f, h). *Il18r1* KO mice (in which IL-18 signaling is absent in the cells in recipient mice, whereas it is intact in transferred ex-T cells) or IL-18 neutralizing antibody (which blocks IL-18 signaling) were used to abrogate the function of IL-18Rα[hi] en-T or en-T/ex-

T cells (Fig. 5i). The use of *Il18r1* KO mice or anti-IL-18 treatment did not result in significant differences in tumor growth rate and survival among CTX[pre]-experienced groups (WT-CTX[pre], WT-CTX[pre] + αIL-18, *Il18r1* KO-CTX[pre]; Fig. 5j, k), which had low proportion of IL-18Rα[hi] T cells. Importantly, CTX[pre]/CD4[post] groups showed contrasting results between endogenous cell-restricted and whole-cell blocking of IL-18

**Fig. 5 IL-18Rα<sup>hi</sup> CD8+ T-cell-dependent anti-tumor effect is mediated by TCR/IL-18 signaling. a** Polyfunctionality of IL-18Rα$^{low}$, IL-18Rα$^{mid}$, and IL-18Rα$^{hi}$ endogenous CD8+ T (en-T) cells was calculated as in Fig. 3b using the data from Supplementary Fig. 5c, d. Analytes prepared as in Fig. 4a were used. **b, c** Evaluation of anti-tumor efficacy of IL-18Rα$^{hi}$ en-T cells. **b** Schematic of the experiment. Injected cell number: B16-F10, $2 \times 10^5$/mouse; ex vivo-primed Thy1.1+ Pmel-1 CD8+ T (ex-T) cells, $2 \times 10^6$/mouse; isolated en-T cells (from the 1st transfer mice), $1 \times 10^6$/mouse. **c** Tumor growth until day 32 is shown. Tumor growth curves over the whole period and survival curves are shown in Supplementary Fig. 6a, b. IL-18Rα$^{hi}$ en-T indicates magnetically enriched cells as shown in Supplementary Fig. 3h. Error bars indicate means ± SEM. **d** IL-18Rα expression in ex-T cells was analyzed as in Fig. 4c. Each symbol indicates the calculated value of an individual mouse. Horizontal bars indicate means. **e** Polyfunctionality of IL-18Rα$^{low}$, IL-18Rα$^{mid}$, and IL-18Rα$^{hi}$ ex-T cells was calculated as in Fig. 3b using the data analyzed in Supplementary Fig. 5e, f. **d, e** Analytes prepared as in Fig. 4a were used. **f** Cytokine concentration in culture media was evaluated. Isolated IL-18Rα$^{low/mid}$ and IL-18Rα$^{hi}$ ex-T cells as in Supplementary Fig. 3i were co-cultured with gp100$_{25-33}$-pulsed or unpulsed C57BL/6 splenocytes for 4 h. IL-18 (30 ng/mL) was added to the treatment group. Pooled cells from CTX$^{pre}$/CD4$^{post}$-experienced mice (n = 5) were used in this assay. Biological triplicates. **g, h** Evaluation of anti-tumor efficacy of IL-18Rα$^{hi}$ ex-T cells. **g** Schematic of the experiment. Injected cell number: B16-F10, $2 \times 10^5$/mouse; ex-T cells, $2 \times 10^6$/mouse; isolated ex-T cells (from the 1st transfer mice), $5 \times 10^5$/mouse. **h** Tumor growth until day 32 is shown. Tumor growth curves over the whole period and survival curves are shown in Supplementary Fig. 7a, b. IL-18Rα$^{hi}$ and IL-18Rα$^{low/mid}$ ex-T mean magnetically sorted cells as in Supplementary Fig. 3j. Error bars indicate means ± SD. **i–k** Evaluation of the role of IL-18 signal in CTX$^{pre}$/CD4$^{post}$ regimen. **i** Schematic of the experiment. Injected cell number: B16-F10, $2 \times 10^5$/mouse; ex-T cells, $2 \times 10^6$/mouse. Tumor growth (**j**) and survival rates (**k**) are shown. **j** Each curve indicates an individual mouse. † indicates death at the indicated time point. The number of mice surviving up to day 80 is represented. **g–k** Anti-IL-18 treatment started from day 4 and was performed thrice a week for the whole period. **a–e, i–k** n = 5 mice/group. **g, h** WT groups, n = 7 mice/group; *Il18r1* knockout groups, n = 5 mice/group. Two-tailed unpaired Student's t-test (**d**) and two-tailed log-rank (Mantel–Cox) tests (**k**) were used to determine statistical significance. αIL-18, IL-18 neutralizing antibody; αMHC-1, anti-MHC-1 blocking antibody; CD4$^{post}$, anti-CD4 post conditioning; CTX$^{pre}$, cyclophosphamide preconditioning; WT, wild-type mice. Source data are provided as a Source Data file.

signal. Anti-IL-18 treatment markedly diminished tumor suppression and survival rate in CTX$^{pre}$/CD4$^{post}$-experienced C57BL/6 wild-type mice. However, CTX$^{pre}$/CD4$^{post}$-experienced *Il18r1* KO mice showed similar therapeutic efficacy as C57BL/6 wild-type mice, implying that the anti-melanoma effect of highly tumor-specific IL-18Rα$^{hi}$ ex-T cells outweigh that of IL-18Rα$^{hi}$ en-T cells. Collectively, these data show that CTX$^{pre}$/CD4$^{post}$ induces a milieu that continuously generates tumor-suppressive IL-18Rα$^{hi}$ CD8+ T cells, which exert strong effector function in response to TCR and IL-18 signaling.

**Various factors in CTX$^{pre}$/CD4$^{post}$ synergistically induce IL-18Rα$^{hi}$ CD8+ T cells.** To determine the variable(s) that regulate the generation of tumor-suppressive IL-18Rα$^{hi}$ ex-T cells, we assessed the roles of factors involved in the current regimen and evaluated the effect of each factor. We treated mice with each factor-subtracted regimen (tumor presence, CTX$^{pre}$, ex-T transfer, IL-2, and CD4$^{post}$ were removed from the full regimen) and analyzed the change in the ratio and number of IL-18Rα$^{hi}$ cells on day 25 (the same experiment scheme as shown in Fig. 4a). In addition, anti-IL-18, *Il18r1* KO, *Foxp3$^{DTR}$*, and *Rag1* KO mice were also analyzed to investigate the effect of IL-18 signaling, regulatory T-cell (Treg) depletion, and permanent lymphopenia. The removal of CTX$^{pre}$, IL-2, and CD4$^{post}$ significantly decreased the frequency and number of IL-18Rα$^{hi}$ ex-T cells (Fig. 6a, b). Notably, the IL-18Rα$^{hi}$ frequency of Treg-depleted mice was similar to that of CD4-depleted mice, suggesting the importance of Treg in the formation of this milieu. However, disruption in IL-18 signaling or Treg depletion reduced the absolute count of IL-18Rα$^{hi}$ ex-T cells compared with that of the control regimen (Fig. 6b). Among the variables, the ratio of IL-18Rα$^{hi}$ subset in en-T cells was significantly affected by CTX$^{pre}$ and CD4$^{post}$ (Supplementary Fig. 8a). The absolute count of IL-18Rα$^{hi}$ en-T cells was strongly dependent on the extent of lymphopenia, probably due to variation in total en-T-cell count at the starting point (Supplementary Fig. 8b, c). Surprisingly, *Rag1* KO mice exhibited the highest frequency of IL-18Rα$^{hi}$ ex-T cells, implying that lymphopenic condition is the important factor contributing to the generation of IL-18Rα$^{hi}$ ex-T cells. This was corroborated by assessing the relationship between the extent of lymphopenia and the ratio of IL-18Rα$^{hi}$ ex-T cells. The frequency of IL-18Rα$^{hi}$ ex-T cells strongly correlated with lymphopenic score (Fig. 6c). En-T cells, which have more complex variables (in priming,

repopulation) than ex-T cells, did not show significant correlation between the variables (Supplementary Fig. 8d).

To understand in more detail the changes induced by CTX$^{pre}$/CD4$^{post}$, we conducted single-cell transcriptome analysis of lymphoid cells. We assessed the differentially abundant cell clusters between regimens (CTX$^{pre}$ and CTX$^{pre}$/CD4$^{post}$) and differentially expressed genes among the identified clusters. In the macroscopic picture of whole lymphoid cells, the addition of CD4$^{post}$ specifically affected CD4+ T cells (Supplementary Fig. 9a), while significant changes were not observed in the other clusters. Next, we subjected en-T and ex-T cells to single-cell transcriptome analysis, to determine the gene expression profile of CD8+ T cells exposed to CTX$^{pre}$/CD4$^{post}$. Compared with the CTX$^{pre}$ group, CTX$^{pre}$/CD4$^{post}$-experienced en-T and ex-T cells had larger size of clusters that expressed high level of *Il18r1* (Fig. 6d and Supplementary Fig. 9b). In line with the previous result (Fig. 4f), these clusters also showed dominant expression of several genes that were highly expressed in the IL-18Rα$^{hi}$-enriched en-T cells (Fig. 6e and Supplementary Fig. 9c). As lymphopenic condition, in which various cytokine signals are amplified[29–31], was the key factor involved in the generation of IL-18Rα$^{hi}$-enriched ex-T cells (Fig. 6a–c), we hypothesized that the cytokine cues from CTX$^{pre}$/CD4$^{post}$-induced lymphopenic milieu affect the expansion of IL-18Rα$^{hi}$ ex-T cells. Therefore, we determined the expression levels of genes related to receptors of homeostatic/inflammatory cytokines. Interestingly, the cell cluster with high *Il18r1* level also showed higher expression of *Il7r* compared with *Il12rb1*, *Il12rb2*, *Il15ra*, and *Il21r* (Fig. 6f and Supplementary Fig. 9d), implying that IL-7 signal can affect IL-18Rα$^{hi}$ cell generation. To verify this, IL-7-Fc treatment was introduced during the current CTX$^{pre}$/CD4$^{post}$ regimen (Fig. 6g). Strikingly, the amplification of the IL-7 signal profoundly increased the number and ratio of IL-18Rα$^{hi}$ en-T and ex-T cells (Fig. 6h and Supplementary Fig. 10).

To summarize, these results suggest that CTX$^{pre}$/CD4$^{post}$-created lymphopenic condition, which involves amplification of the homeostatic IL-7 signal, increases IL-18Rα$^{hi}$ CD8+ T-cell subset.

## Discussion

ACT efficacy is often limited by immune-suppressive elements (e.g., Tregs and myeloid-derived suppresser cells)[32,33]. To resolve this issue, patients are treated with suitable pre- and/or post-conditioning regimen. Pre-conditioning includes radiation and

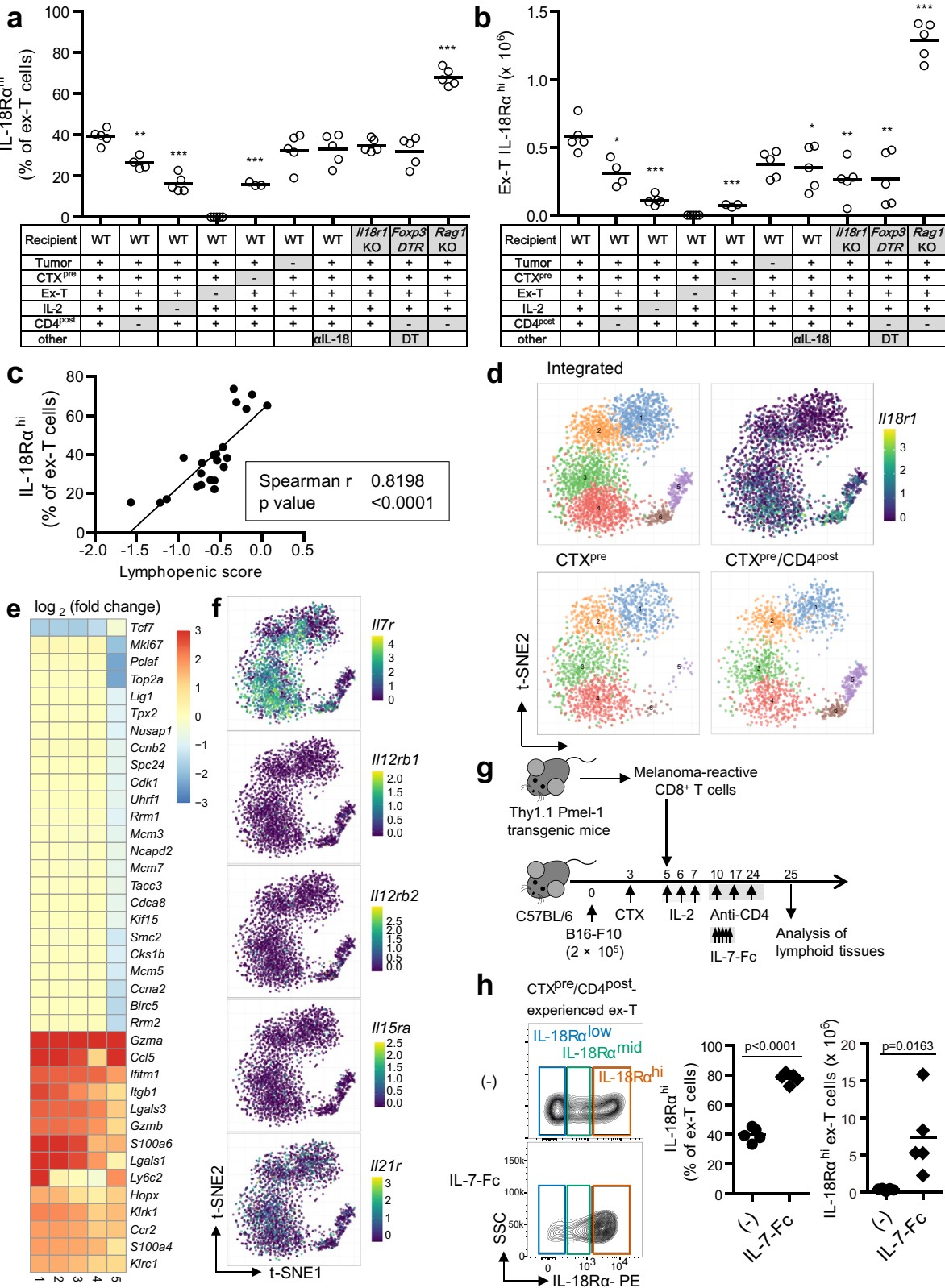

chemo-agent conditioning that induce lymphodepletion, thereby eliminating consumptive cytokine sinks and immune-suppressive cells[12–14]. This process also creates "space" for the homeostatic expansion of ex-T cells[34,35]. In addition, post conditioning complements supportive elements (e.g., homeostatic cytokines) that allow the long-term persistence of ex-T cells[36,37]. In the current study, we designed CTX$^{pre}$/CD4$^{post}$, a regimen that

transiently depletes CD4$^+$ cells following CTX-induced systemic lymphodepletion. In our mouse model of ACT for B16-F10 melanoma, CTX$^{pre}$/CD4$^{post}$ increased anti-tumor efficacy by accelerating ex-T-cell expansion and en-T-cell differentiation. Notably, CTX$^{pre}$/CD4$^{post}$-exposed ex-T and en-T cells heightened the frequency of IL-18Rα$^{hi}$ subset, which was highly polyfunctional and exhibited TCR/IL-18 signal-dependent anti-

**Fig. 6 Various factors affect IL-18Rα$^{hi}$ cell generation in CTX$^{pre}$/CD4$^{post}$-induced milieu. a–c** Evaluation of variables that led to IL-18Rα$^{hi}$ cell generation. The treatment with full regimen (as in Fig. 4a) or a modified regimen that lacks each variable was applied to mice ($n = 5$ mice per group). On day 25, the ratio (**a**) and the number (**b**) of IL-18Rα$^{hi}$ Thy1.1$^{+}$ Pmel-1 CD8$^{+}$ T (ex-T) cells in lymphoid tissues were analyzed. **c** The correlation between lymphopenia score (represented as −log [total cell count in spleen and tumor-draining lymph nodes]) and the proportion of IL-18Rα$^{hi}$ ex-T cells was evaluated. **d–f** Visualization of single-cell transcriptome analysis of CTX$^{pre}$ and CTX$^{pre}$/CD4$^{post}$-experienced ex-T cells. Cells from four mice of each group were pooled and analyzed. **d** After unsupervised clustering, the ex-T cells were projected on two-dimensional space using *t*-distributed stochastic neighbor embedding (t-SNE). The information related to groups, clusters, and *Il18r1* expression are shown. **e** Genes that are differentially expressed compared with those in the cluster 6 are listed. **f** The log-normalized transcript levels of the indicated genes are shown. **g, h** Evaluation of the effect of IL-7 signal on IL-18Rα$^{hi}$ CD8$^{+}$ T-cell generation. **g** Schematic of the experiment. Injected cell number: B16-F10, $2 \times 10^{5}$/mouse; ex-T cells, $2 \times 10^{6}$/mouse. The treatment group received daily treatment of non-lytic Fc-fused IL-7 (IL-7-Fc) from day 10–14. **h** Representative data, the ratio, and the number of IL-18Rα$^{hi}$ ex-T cells are shown. **a, b, h** $n = 5$ mice per group. Each symbol indicates the calculated value of an individual mouse. Horizontal bars indicate means. *$p < 0.05$; **$p < 0.01$; ***$p < 0.001$; one-way ANOVA with the Dunnett's post hoc test (**a, b**; the first group was the control), two-tailed Spearman's rank correlation (**c**), and two-tailed unpaired Student's *t*-test (**h**) were used to determine statistical significance. αIL-18, IL-18 neutralizing antibody; CD4$^{post}$, anti-CD4 post conditioning; CTX$^{pre}$, cyclophosphamide preconditioning; DT, diphtheria toxin; WT, wild-type mice. Source data are provided as a Source Data file.

---

tumor activity in vivo. Despite the short-term effect of IL-18Rα$^{hi}$ CD8$^{+}$ T cells in adoptive transfer experiments, the CTX$^{pre}$/CD4$^{post}$-created milieu, which continuously generates the IL-18Rα$^{hi}$ subset, elicited profound anti-tumor efficacy in an IL-18 signal-dependent manner. We also found that IL-18Rα$^{hi}$ CD8$^{+}$ T-cell generation is strongly associated with the existence of both CTX$^{pre}$ and CD4$^{post}$, or lymphopenic milieu such as in *Rag1* KO mice. Single-cell transcriptome analysis and subsequent experiments revealed that the amplification of IL-7 signaling was crucial for increasing IL-18Rα$^{hi}$ ex-T and en-T cells.

The therapeutic action of the anti-CD4-depleting antibody in cancer, in particular via reduction in Tregs, is well known[5,15]. As Tregs suppress the adequate function and proliferation of effector T cells[38], CD4$^{+}$ Treg depletion increases tumor reactivity and intratumoral infiltration of CD8$^{+}$ T cells[3–5]. In the present study, we showed that anti-CD4 treatment, in combination with ACT and CTX$^{pre}$, enhanced anti-melanoma efficacy by enriching IL-18Rα$^{hi}$ CD8$^{+}$ T cells. Notably, the synergy between ACT (i.e, ex-T) and anti-CD4 was crucial in this study, as the removal of ex-T compromised the therapeutic effect (Fig. 1c–f). This was further corroborated by the robust anti-tumor response of CTX$^{pre}$/CD4$^{post}$-experienced *Il18r1* KO mice (Fig. 5j, k), in which the highly functional IL-18Rα$^{hi}$ subset is only present in ex-T, but not in en-T cells. Interestingly, the depletion of Tregs, instead of whole CD4$^{+}$ cells, also increased IL-18Rα$^{hi}$ ex-T-cell frequency (Fig. 6a). However, the therapeutic effect of Treg depletion was overshadowed by its side effects. Treg-depleted mice (diphtheria toxin (DT)-treated *Foxp3$^{DTR}$*) developed typical autoimmune symptoms and died within a month (Supplementary Fig. 11), as has been observed in other studies[39,40]. In contrast, the regimen used in this study (weekly treatment of anti-CD4) did not elicit any autoimmune symptoms, while the anti-tumor response was retained. This is possibly due to the collateral removal of Th subsets with Tregs, as Th1, Th9, and Th17 play important roles in autoimmune diseases[41,42]. One concern is the elimination of tumor-suppressive CD4$^{+}$ T cells, as these cells are known for their roles in direct tumor suppression and T-cell memory formation[43–48]. All these effects were removed by the CTX$^{pre}$/CD4$^{post}$ regimen in this study. The optimal level and duration of CD4 depletion should be considered in the future to maximize therapeutic efficacy by balancing these factors.

In this study, anti-tumor efficacy in CTX$^{pre}$/CD4$^{post}$-experienced mice was largely attributed to the IL-18Rα$^{hi}$ CD8$^{+}$ subset. This is in line with the observations of previous studies that demonstrated elevated effector function of IL-18Rα$^{hi}$ CD8$^{+}$ T cells in infection and cancer[28,49]. We found that the IL-18Rα$^{hi}$ subset had highly differentiated effector profile with skewed TCR repertoire (Fig. 4e, f). Despite the short-term effector function, the cell subset was polyfunctional and required TCR–MHC I

interaction for anti-tumor activity (Fig. 5a, c, e, f). Notably, IL-18 signaling was crucial in TCR-mediated effector function of IL-18Rα$^{hi}$ ex-T cells in vitro and in vivo (Fig. 5f, h, j, k). In the case of CTX$^{pre}$-experienced mice, which had lower capacity of generating the IL-18Rα$^{hi}$ subset, inhibition of IL-18 signaling did not significantly change anti-tumor activity (Fig. 5j, k). In summary, IL-18Rα$^{hi}$ CD8$^{+}$ T cells are tumor-specific and act as short-term effector cells that drive IL-18-dependent therapeutic effect under CTX$^{pre}$/CD4$^{post}$.

The significance of this study lies in the design of the CTX$^{pre}$/CD4$^{post}$ regimen that creates a favorable milieu for effective ACT. Anti-CD4 treatment, when combined with CTX$^{pre}$, increased the frequency of IL-18Rα$^{hi}$ ex-T and en-T cells (Fig. 6a and Supplementary Fig. 8a). We also found that lymphopenia score correlated strongly with the proportion of IL-18Rα$^{hi}$ ex-T cells (Fig. 6c). Considering that Napolitano et al.[29] and Malaspina et al.[50] reported elevation in IL-7 level in hosts harboring human immunodeficiency virus 1 and showing idiopathic CD4$^{+}$ T-cell decrease, it is possible that CTX$^{pre}$/CD4$^{post}$-induced IL-7 abundance contributes to IL-18Rα$^{hi}$ cell generation. This was confirmed in this study by showing that IL-7-Fc significantly increased the IL-18Rα$^{hi}$ cell population that expressed high level of *Il7r* (Fig. 6f, h and Supplementary Figs. 9d and 10). In addition, CTX$^{pre}$/CD4$^{post}$ also contributes to the function of the cells via abundant IL-18 signaling, as pro-inflammatory cytokines (as in the case of homeostatic cytokines) are also abundant in lymphopenic hosts[29–31,50]. Therefore, CTX$^{pre}$/CD4$^{post}$ is involved in both the generation and function of the IL-18Rα$^{hi}$ subset via augmented homeostatic and pro-inflammatory signals. Collectively, the merit of the CTX$^{pre}$/CD4$^{post}$ regimen is, as can be expected from its mechanism of action, that it can be applied for any type of ACT (e.g., sorted CD8$^{+}$ T cells, TILs, TCR-engineered T cells, and CAR-Ts). However, this effect was evaluated only in the mouse model of melanoma, which is a limitation of this study. Further studies of other tumor models and translation into the human setting will pave the way for developing safe and effective ACT strategies.

## Methods

**Study design**. This study was conducted to develop an effective conditioning regimen in ACT. In general, the B16-F10 melanoma cell line was inoculated into animals and the CD8$^{+}$ T cells were adoptively transferred. Animals in two different groups were treated with either CTX or CTX plus anti-CD4, to compare the effects of conditioning regimens. Animals were evaluated for tumor growth and survival, and both the transferred and endogenous CD8$^{+}$ T cells were analyzed to investigate the mechanisms underlying any relevant outcome. Sample sizes were selected based on previous experience with similar types of experiments. The study was conducted with multiple technical and/or biological replicates, to ensure reproducibility of data. In all experiments, animals were randomly assigned into experimental groups and non-blinded experiments were conducted throughout the study. No data were excluded and all outliers were included in data analysis.

**Cells.** B16-F10 cells purchased from the American Type Culture Collection were grown in complete Dulbecco's modified Eagle's medium (D10; Welgene, Gyeongsan, Korea; supplemented with 10% heat-inactivated fetal bovine serum [FBS], 100 units/mL penicillin, and 100 μg/mL streptomycin). Lymphocytes derived from animals were cultured in Roswell Park Memorial Institute 1640 medium (R10; Welgene, Gyeongsan, Korea; supplemented with 10% heat-inactivated FBS, 100 units/mL penicillin, and 100 μg/mL streptomycin). All cells were cultured at 37 °C in a 5% $CO_2$ incubator.

**Mice.** Six- to 8-week-old C57BL/6 female mice were purchased from OrientBio (Gapyeong, Korea). CD45.1 congenic (B6.SJL-$Ptprc^a$ $Pepc^b$/BoyJ), $Rag1$ KO ($Rag1^{tm1Mom}$/J), $Il18r1$ KO (B6.129P2-$Il18r1^{tm1Aki}$/J), $Foxp3^{DTR}$ (B6.129(Cg)-$Foxp3^{tm3(DTR/GFP)Ayr}$/J), and Thy1.1 Pmel-1 transgenic mice ($Thy1^a$/Cy Tg(TcraTcrb)8Rest/J; expressing transgenic TCR specific for gp100$_{25-33}$ and congenic Thy1.1 antigen) were purchased from the Jackson Laboratory (Bar Harbor, ME, USA). All mice were maintained under specific pathogen-free conditions in the animal facilities of the National Cancer Center (Goyang, Korea). Groups of up to five mice were housed in individually ventilated cage on a 12 h light/dark cycle, at an ambient temperature of 20 ± 2 °C, with humidity controlled at 55 ± 5%. Experimental procedures were approved by the Institutional Animal Care and Use Committee of the National Cancer Center Institute (NCCI). The NCCI animal facility is fully accredited by the Association for Assessment and Accreditation of Laboratory Animal Care International. Animal experiments were conducted following the Guidelines on the Care and Use of Laboratory Animals from the Institute of Laboratory Animal Resources.

**Sample preparation.** To prepare Thy1.1$^+$ Pmel-1-specific and CD45.1$^+$ CD8$^+$ T cells, lymphocytes were collected from the lymph nodes and spleens of Thy1.1 Pmel-1 and CD45.1 mice, respectively. CD8$^+$ T cells were isolated from lymphocytes using CD8 microbeads (positive selection beads; Miltenyi Biotech, CA, USA). For ex vivo priming, the Thy1.1$^+$ Pmel-1 CD8$^+$ T cells were further stimulated with 5 μg/mL human gp100$_{25-33}$ peptide (KVPRNQDWL; Peptron, Daejeon, Korea) in R10 media, in the presence of 5% non-irradiated splenocytes for 2 days. The lymph nodes and spleen were gently disrupted and filtered through a 40 μm nylon cell strainer (Falcon, NY, USA) to make single-cell suspensions. Cells were treated with red blood cell lysis buffer (eBioscience, CA, USA) before use.

To study the characteristics and functions of en-T/ex-T cells in adoptively transferred mice, the mixed cells in tumor-draining (inguinal) lymph nodes and spleen were used if not otherwise specified. For functional analysis, cells were isolated using a separate procedure depending on their purpose. Thy1.1 and CD8 microbeads (positive selection beads; Miltenyi Biotec, Inc., CA, USA) were used for in vitro killing assay (as in Fig. 3d and see Supplementary Fig. 3e). Fluorescence-activated cell sorting (FACSAria, BD Biosciences, NJ, USA) was performed for TCR clonotyping and microarray analyses (as in Figs. 3e and 4b, and see Supplementary Fig. 3f). To isolate IL-18Rα$^{hi}$-enriched en-T cells for TCR clonotyping and bulk RNA sequencing (as in Fig. 4e, f), Thy1.1$^-$ CD8$^+$ T cells were isolated from lymphoid tissues of mice with a CD8 isolation kit (negative selection; Miltenyi Biotec, Inc., CA, USA) and Thy1.1 microbeads. Then, the IL-18Rα$^{hi}$-enriched cells were positively selected with phycoerythrin (PE)-labeled IL-18Rα antibody (eBioscience, CA, USA), PE microbeads (Miltenyi Biotec, Inc., CA, USA), and LS columns (positive selection; Miltenyi Biotec, Inc., CA, USA). The en-T cells that passed through the column were collected and used as a control for comparison with IL-18Rα$^{hi}$-enriched en-T cells (Supplementary Fig. 3g). For adoptive transfer of IL-18Rα$^{hi}$ en-T cells (as in Fig. 5b, c), Thy1.1$^+$ CD8$^+$ T cells were isolated using a CD8 isolation kit (negative selection) and Thy1.1 microbeads. Then, the IL-18Rα$^{hi}$ cells were positively selected with PE-labeled IL-18Rα antibody, PE microbeads, and LS columns (Supplementary Fig. 3h). Fluorescence-activated cell sorting (FACSMelody, BD Biosciences, NJ, USA) was performed (Supplementary Fig. 3i) for assaying in vitro cytokine secretion (as in Fig. 5f). For adoptive transfer of IL-18Rα$^{hi}$ and IL-18Rα$^{low/mid}$ ex-T cells (as in Fig. 5g, h), Thy1.1$^+$ CD8$^+$ T cells were isolated via negative selection using a CD8 isolation kit (negative selection) and Thy1.2 microbeads (Miltenyi Biotec, Inc., CA, USA). IL-18Rα$^{hi}$ and IL-18Rα$^{low/mid}$ cells were separated using PE-labeled IL-18Rα antibody, PE microbeads, and LS columns (Supplementary Fig. 3j).

Tumor tissue-derived cells were prepared using a tumor dissociation kit (Miltenyi Biotec, Inc., CA, USA) according to the manufacturer's instructions. Briefly, tumor tissues collected from mice were cut into small pieces and transferred to gentleMACS C tubes containing an enzyme mix. The mixture was processed using a gentleMACS dissociator (Miltenyi Biotec, Inc., CA, USA) and filtered through a 40 μm nylon cell strainer to obtain single-cell suspensions.

**Adoptive transfer models.** The adoptive transfer experiment was conducted throughout this study. On day 0, C57BL/6, $Rag1$ KO, $Il18r1$ KO, or $Foxp3^{DTR}$ mice were intradermally injected with $2 \times 10^5$ B16-F10 cells. Alternatively, for analyzing tumor tissue (Fig. 3f–i), mice were intradermally injected with $2 \times 10^6$ B16-F10 cells to establish analyzable tumor volume in all experimental groups. On the day of the injection, B16-F10 cells that showed 90% confluence and >80% viability (checked using ADAM-MC2 [NanoEnTek, Seoul, Korea]) were used. On day 3, preconditioning lymphodepletion was induced via intraperitoneal injection of

300 mg/kg CTX (Baxter Oncology GmbH, Halle/Westfalen, Germany). On day 5, mice received intravenous injections of CD8$^+$ T cells via the lateral tail vein. In addition, starting on day 5, recombinant human IL-2 (10,000 IU; Novartis, Basel, Switzerland) was intraperitoneally injected daily for 3 days. Starting from day 10, anti-CD4 post conditioning involved weekly intraperitoneal injections of 200 μg anti-mouse CD4 monoclonal antibody (clone: GK1.5; BioXCell, NH, USA) for up to 5 weeks. To block MHC I–TCR interaction, 250 μg anti-mouse MHC class I (H-2) antibody (clone:M1/42.3.9.8; BioXCell) was administered every 3 days until the entire duration of the experiment. To inhibit IL-18 signaling, 200 μg anti-mouse IL-18 monoclonal antibody (clone: YIGIF74-1G7; BioXCell) was intraperitoneally administered thrice a week from day 4 until the entire duration of the experiment. Treg depletion was performed by injecting $Foxp3^{DTR}$ mice 10 μg/kg DT (Sigma–Aldrich, MO, USA) intraperitoneally for every 2 days from day 10 until the entire duration of the experiment. For increasing IL-7 signaling, non-lytic IL-7-Fc (1 μg; Adipogen, CA, USA) was daily injected intraperitoneally for 5 days until day 10.

For secondary adoptive transfer, en-T or ex-T cells collected from the first-recipient mice (C57BL/6 mice that experienced the general CTX$^{pre}$/ACT/CD4$^{post}$ regimen as in Fig. 4d) were adoptively transferred to the second-recipient mice. To compare the anti-tumor effect of en-T and IL-18Rα$^{hi}$ en-T cells (Fig. 5b), the second-recipient $Rag1$ KO mice were intradermally injected with $2 \times 10^5$ B16-F10 cells on day 0, followed by intraperitoneal injection of 300 mg/kg CTX on day 3. On day 5, $1 \times 10^6$ en-T or IL-18Rα$^{hi}$ en-T cells that were isolated from the first-recipient mice (as in Supplementary Fig. 3h) were transferred to B16-F10-bearing $Rag1$ KO mice. Recombinant human IL-2 (10,000 IU) was intraperitoneally injected daily from day 5 to 7. To compare the anti-tumor effect of IL-18Rα$^{hi}$ and IL-18Rα$^{low/mid}$ ex-T cells (Fig. 5g), the second-recipient C57BL/6 or $Il18r1$ KO mice were intradermally injected with $2 \times 10^5$ B16-F10 cells on day 0, followed by an intraperitoneal injection of 300 mg/kg CTX on day 3. On day 5, $5 \times 10^5$ IL-18Rα$^{hi}$ or IL-18Rα$^{low/mid}$ ex-T cells that were isolated from the first-recipient mice (as in Supplementary Fig. 3j) were transferred to B16-F10-bearing C57BL/6 or $Il18r1$ KO mice. Recombinant human IL-2 (10,000 IU) was intraperitoneally injected daily from day 5 to 7.

In vivo CD8$^+$ T-cell proliferation was evaluated according to the following procedure (Fig. 2e). On day 0, C57BL/6 mice were intradermally injected with phosphate-buffered saline or $2 \times 10^5$ B16-F10 cells. On day 3, lymphodepletion was induced via intraperitoneal injections of 300 mg/kg CTX. Activated Thy1.1$^+$ Pmel-1 and unstimulated CD45.1$^+$ CD8$^+$ T cells were labeled with 10 μM carboxyfluorescein succinimidyl ester and violet fluorescence using a CellTrace CFSE cell proliferation kit (Thermo Fisher Scientific, MA, USA) and a CellTrace Violet cell proliferation kit (Thermo Fisher Scientific, MA, USA), respectively. On day 5, the two cell populations were mixed in 1 : 1 ratio and transferred to B16-F10-bearing C57BL/6 mice ($2 \times 10^6$ cells per mice). Starting from day 5, IL-2 was intraperitoneally injected daily for 3 days. Starting from day 10, mice were subjected to weekly intraperitoneal injections of 200 μg anti-mouse CD4 antibody. CD8$^+$ T-cell proliferation was analyzed on day 25.

Mice were routinely monitored for tumor growth and survival. Tumor volume was calculated as: $1/2 \times$ length $\times$ width $\times$ height. Mice with a tumor volume >2500 mm$^3$ were killed as per guidelines.

**Flow cytometry.** To analyze the expression of general surface markers, single-cell suspensions were incubated with antibody to block the CD16 and CD32 receptors (2.4G2) for 10 min at 4 °C. The cells were then stained either with Fixable Viability Dye eFluor 780 (eBioscience, CA, USA) or Fixable Viability Stain 700 (BD Biosciences, NJ, USA) and then stained with marker-specific antibodies for 30 min at 4 °C in the dark. The stained cells were washed, fixed with 1% paraformaldehyde, and stored at 4 °C in the dark until analysis.

To evaluate intracellular cytokine expression, cells were placed in the well ($5 \times 10^5$/well) of a 96-well round-bottom plate, treated with a Cell Stimulation Cocktail and Protein Transport Inhibitor Cocktail (Thermo Fisher Scientific, MA, USA), and then incubated for 6 h. Subsequently, the cells were washed and stained with fluorochrome-conjugated antibodies using a Cytofix/Cytoperm Fixation/Permeabilization Solution Kit (BD Biosciences, NJ, USA).

Flow cytometry was performed using FACSVerse and LSRFortessa (BD Biosciences, NJ, USA). The data were analyzed in FlowJo (Tree Star, Inc., OR, USA). All gating strategies and representative data in this study are indicated in Supplementary Figs. 2, 3, and 5. The t-distributed stochastic neighbor embedding plugin in FlowJo was used to visualize multi-color flow cytometry data set (Fig. 4g). The data set was analyzed after concatenation of the same number of en-T cells from each experimental group. The fluorochrome-labeled antibodies used in this study are listed in Supplementary Table 1.

**In vitro functional assays.** To evaluate tumor reactivity (as in Figs. 3c), $1 \times 10^6$ lymphocytes were added to a 48-well plate and cultured in the absence or presence of $1 \times 10^5$ B16-F10 cells for 16 h. After incubation, the cells were stained with fluorochrome-conjugated anti-CD25, anti-CD8, and anti-Thy1.1 antibodies before analysis.

The in vitro killing assay (Fig. 3d) began with the seeding of $1 \times 10^4$ B16-F10 cells in 96-well clear-bottom black plates. After 2 days, the B16-F10 cell count per well was determined by calculating the average cell number of three representative

wells. The en-T cells and seeded B16-F10 cells were co-cultured at ratios of 1 : 3, 1 : 1, 3 : 1, 9 : 1, or 27 : 1 for 18 h. The extent of B16-F10 cell death was measured using a CytoTox-Glo cytotoxicity assay kit (Promega, WI, USA) and TECAN infinite PRO 200 (Tecan Group, Männedorf, Switzerland). The killing efficacy of en-T cells was calculated as follows: (cytotoxicity signal in sample well − background cytotoxicity signal)/maximum cytotoxicity signal). Background and maximum signals were measured in wells containing B16-F10 cells alone and in wells with digitonin-treated B16-F10 cells, respectively.

To analyze in vitro cytokine secretion (as in Fig. 5f), the isolated IL-18Rα^hi and IL-18Rα^low/mid ex-T cells ($1 \times 10^4$/well) were added to a 96-well plate and co-cultured with C57BL/6-derived splenocytes ($1 \times 10^4$/well) for 4 h. Depending on the assay groups, splenocytes that were unpulsed or pulsed with 5 μg/mL human gp100_{25–33} peptide for 1 h were used. For IL-18 treatment group, 30 ng/mL recombinant mouse IL-18 (BioLegend, CA, USA) was applied during the co-culture. After incubation, the cell culture media was analyzed using a CBA mouse Th1/Th2/Th17 cytokine kit (BD Biosciences, NJ, USA) and flow cytometry.

**TCR analysis.** Total RNA of lymphocytes ($1 \sim 5 \times 10^6$) was extracted using the RNeasy Plus mini kit (Qiagen, Hilden, Germany). To generate the TCR-sequencing library, 100 ng RNA per sample was reverse transcribed and amplified by nested PCR with a SMARTer mouse TCR a/b profiling kit (Takara Bio, CA, USA). Illumina-specific adaptor sequences were incorporated during amplification. The TCR-sequencing library was size-selected and purified using AMPure XP beads (Beckman Coulter, IN, USA). Final libraries were validated using Agilent 2100 Bioanalyzer and a DNA 1000 kit (Agilent, CA, USA) before sequencing in an Illumina MiSeq sequencer with paired-end, $2 \times 300$ base pair reads (Illumina, CA, USA). Adapter sequences were removed from raw data using Scythe (v0.994) and Sickle. Sequences were cleaned by excluding reads shorter than 36 base pairs. The processed data were then aligned in MiXCR (https://mixcr.readthedocs.io). Aligned TCRα/β sequences were extracted using VDJtools (https://vdjtools-doc.readthedocs.io). The raw TCR read ranged as follows: α-chain, 380,844–638,615 (Fig. 3e); β-chain, 762,389–1,028,175 (Fig. 3e); α-chain, 222,916–499,238 (Fig. 4e); and β-chain, 335,371–736,419 (Fig. 4e). All TCR reads were down-sampled to 10,000 reads per sample and used for computing TCR diversity with the CalcDiversityStats routine. Evenness of TCR diversity was determined using the normalized Shannon–Wiener index.

**Gene expression analysis.** For microarray analysis (Fig. 4b), total RNA was extracted from endogenous Thy1.1⁻CD8⁺ T cells pooled from five ACT-treated mice using Trizol reagent (Invitrogen, CA, USA). RNA purity and integrity were evaluated using a ND-1000 spectrophotometer (NanoDrop, Wilmington, USA) and Agilent 2100 Bioanalyzer (Agilent Technologies, Palo Alto, CA, USA), respectively. The Affymetrix Whole Transcript Expression array was used following the manufacturer's protocol (GeneChip Whole Transcript PLUS reagent kit). cDNA was synthesized following the manufacturer's protocol using the GeneChip Whole Transcript Amplification kit. The sense cDNA was then fragmented and biotin-labeled with terminal deoxynucleotidyl transferase using a GeneChip Whole Transcript Terminal Labeling kit. Approximately 5.5 μg labeled DNA was hybridized to the Affymetrix GeneChip Clariom S mouse array for 16 h at 45 °C. The hybridized arrays were washed and stained on a GeneChip Fluidics Station 450 and then scanned with a GCS3000 Scanner (Affymetrix). Array data export processing and analysis was performed using Affymetrix GeneChip Command Console Software. Data were normalized with a robust multi-average method in Affymetrix Power Tools.

For whole transcriptome analysis (Fig. 4f), total RNA was extracted from IL-18Rα^hi-enriched endogenous CD8⁺ T cells and the residual cell population using Trizol reagent. RNA purity and integrity were evaluated using a ND-1000 spectrophotometer and Agilent 2100 Bioanalyzer, respectively. cDNA libraries were prepared for 150 bp paired-end sequencing using TruSeq Stranded mRNA sample preparation kit (Illumina, CA, USA). In brief, the mRNA molecules were purified and fragmented from 1 μg total RNA using oligo (dT) magnetic beads. The fragmented mRNAs were synthesized as single-stranded cDNAs using random hexamer primers. Double-stranded cDNA was prepared using this as a template for second-strand synthesis. After sequential end repair, A-tailing, and adapter ligation, the cDNA libraries were amplified via PCR. The quality of these cDNA libraries was evaluated using the Agilent 2100 BioAnalyzer (Agilent, CA, USA) and they were quantified using the KAPA library quantification kit (Kapa Biosystems, MA, USA) according to the manufacturer's protocol. Following cluster amplification of denatured templates, paired-end ($2 \times 150$ bp) sequencing was performed using Illumina NovaSeq 6000 sequencer (Illumina, CA, USA). Low-quality reads were filtered according to the following criteria: reads containing >10% of skipped bases (marked as 'N's), reads containing >40% of bases whose quality scores were <20, and reads with average quality score <20. The whole filtering process was performed using the in-house scripts (TheragenEtex BiO Institute, Suwon, Korea). The filtered reads were mapped to the reference genome (mm10) using the aligner TopHat. Gene expression level was determined using Cufflinks v2.1.1. To improve the accuracy of the measurement, multi-read-correction and frag-bias-correct options were applied. All other options were set to default values. Genes fitting the following criteria were listed and visualized using Morpheus (Broad Institute); fragments per kilobase of transcript per million mapped reads (FPKM) ≥ 5 in IL-18Rα^hi-enriched group, $p$-value < 0.001, $q$-value < 0.001, mean FPKM of IL-18Rα^hi-enriched group ≥ 3 × mean FPKM of control group, FPKM of each sample in IL-18Rα^hi-enriched group ≥ mean FPKM of control group + 3 × STD of control group.

For single-cell transcriptome analysis (Fig. 6d–f and Supplementary Fig. 9, immune cells were collected from C57BL/6 mice that experienced the general CTX^pre/ACT regimen with and without CD4^post as in Fig. 4a. The following cell fractions were prepared using the cells from the spleen and tumor-draining lymph nodes: whole lymphoid cells, en-T and ex-T cells (magnetically sorted as in Supplementary Fig. 3e). The CTX^pre or CTX^pre/CD4^post-experienced lymphoid cells, en-T, and ex-T cells were labeled using a BD mouse immune single-cell multiplexing kit (BD Biosciences, NJ, USA) and subjected to cDNA library construction using the BD Rhapsody system, a BD Rhapsody WTA amplification kit, and a BD Rhapsody cDNA kit (BD Biosciences, NJ, USA). Paired-end ($2 \times 150$ bp) sequencing was performed using the Illumina HiSeq 2500 sequencer (Illumina, CA, USA). Following sequencing, data processing was performed by running the BD Rhapsody Analysis pipelines for sequencing on the Seven Bridges Genomics platform and the raw data were converted into a matrix containing the values of molecules per cell. R version 4.0.3 and batchelor, BiocSingular, edgeR, pheatmap, SingleCellExperiment, scater, scran, and uwot were used to analyze and visualize the matrix.

**Statistics and reproducibility.** All statistical data were analyzed in Prism v5.01 GraphPad (CA, USA). Two-tailed unpaired Student's $t$-test was used to compare cell counts and frequencies. Two-way analysis of variance (ANOVA) and Bonferroni post hoc test were used for the statistical analysis of in vitro killing assay and tumor growth (Fig. 3d and Supplementary Figs. 6c, 7c, and 11b). One-way ANOVA with the Dunnett's post hoc test were used to compare the effect of variables against control (Fig. 6a, b and Supplementary Fig. 8a–c). Survival rates were compared with log-rank (Mantel–Cox) tests. Mann–Whitney test was used to compare TCR evenness (Figs. 3e and 4e). Spearman's rank correlation was used to determine correlation. Significance was set at $p < 0.05$.

Experimental data reproduced in at least two independent experiments were used to draw conclusions (with the exception of microarray and next-generation sequencing). The number of repeated experiments is as follows: three times—Figs. 1c–f, 2c, 3b, g–j, 4c, and 5d, and Supplementary Fig. 5b; twice—Figs. 1a, 2b, d, f–j, 3c, d, 4a, e, g, 5a, c, e, f, h, j, k, and 6a–c, h, and Supplementary Figs. 1, 4, 5d, f, 6–8, and 10; once—Figs. 3e, 4b, e, f, and 6d–f, and Supplementary Figs. 9 and 11.

**Reporting summary.** Further information on research design is available in the Nature Research Reporting Summary linked to this article.

## Data availability
The microarray, TCR analysis, bulk, and single-cell RNAseq data generated in this study have been deposited in the GEO database under accession code GSE180291, GSE181280, GSE180439, and GSE180991. The remaining data are available within the Article, Supplementary Information, or Source Data file. Source data are provided with this paper.

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

## Acknowledgements

This work was supported by the National Research Foundation of Korea (NRF) grant funded by the Korean government (2018R1A6A3A01011692 [S.-H.K.] from MOE and 2019R1C1C1008999 [C.H.] from MSIT), National Cancer Center of Korea (NCC-2010190 [C.H.] and NCC-1910050/2110480 [B.K.C.]), and the Korean Ministry of Trade, Industry and Energy (GLOBAL R&D PROJECT, N0000901 [B.S.K.]). We thank the Genomics Core Facility, Flow Cytometry Core Facility, and Animal Sciences Branch at the National Cancer Center Research Institute for technical support.

## Author contributions

S.-H.K. conceived the project and conducted most of the experiments. S.-H.K., C.H. and B.K.C. contributed to the experimental design and subsequent analysis. E.C., Y.I.K. and B.K.C. assisted with the animal experiments. C.H. conducted TCR repertoire analysis and single-cell transcriptome analysis. S.-H.K. and C.H. interpreted the results from TCR repertoire and gene expression analyses. C.H., B.K.C. and B.S.K. supervised the project. S.-H.K., C.H., B.K.C. and B.S.K. wrote the manuscript.

## Competing interests

B.S.K. is the founder and Chief Executive Officer of Eutilex Co., Ltd. The remaining authors declare no competing interests.
