## [Peer Review File · Nature Communications]

Reviewers' Comments:

Reviewer #1:

Remarks to the Author:

This paper develops an experimental model of tumor rejection using an adoptive transfer of ex-vivo activated tumor specific CD8 T cells associated with pre-transfer lympho-depletion followed by IL-2 treatment and CD4 depletion. CD4 depletion on the top of lympho-depletion leads to a better anti-tumor effect which is attributed to the expansion of "innate-like" CD8 T cells. These IL-18Ra high CD8 T cells are characterized at the phenotypic and transcriptional level. These IL-18-Ra high CD8 T cells encompasses NKT, MAIT and "bystander" memory T cells.

Although the model is complex and artificial, on a whole, the work is interesting but several conclusions are not fully supported by the data. The phenotypic, lymphokine expression and transcriptome characterization of the different subsets is correctly performed but does not provide real new insight besides the fact that cells that have extensively proliferated display effector or memory features with the corresponding phenotypic and functional properties.

Most experiments seem to have been performed only once (with 4-5 data points each) and the number of mice is often very limited. This is a very serious issue. Science is about reproducibility and every experiment should be performed at least twice. This applies also to the mouse experiments. Two groups in fig. 1C included 10 mice while the number of mice is much lower in the other experiments. It is nowhere stated that the experiment was reproducible (5 mice twice is better than 10 mice once).

The key experiment displayed in 5F is not only performed only once but also does not contain the positive control performed in 4B. This is incorrect. Comparison between experiments is deduction but not a demonstration.

It is not clear whether the experimental model is representative of the human situation in which fludarabine is often used and leads to a long-term CD4 depletion. The model is complex with at least 6 variables

- In vitro primed tumor reactive CD8 T cells
- Lympho depletion
- IL-2
- CD4 depletion (decreasing both Tregs and conventional CD4)
- Endogenous conventional CD8 T cells
- Endogenous IL-18-Ra high CD8 T cells

How these variables interact with each other is unclear and the authors focus mainly on the CD4 T cell depletion but other parameters may be involved. How CD4 depletion leads to the inflation of IL-18-Ra high CD8 T cells is not deciphered. Is injection of IL-2 necessary? This can easily be tested by doing the experiment without IL-2 injection. Is Treg depletion involved? This can be tested by using FoxP3-DTR mice. Is there any role for the tumor specific T cells? This can be tested by not doing this transfer.

Another concern is about the antigen specificity of the IL-18-Ra high CD8 T cells and the mechanisms of their involvement in tumor rejection. The authors observe an important inflation of IL-18-Ra high CD8 T cells which appear to be a heterogeneous subset with regards to antigen specificity. This is not really deciphered (no CD1d/aGC or MR1/5-OP-RU tetramer staining) and no use of the available CD1d or MR1 KO mice. The so-called bystander CD8 T cells are not analyzed further. Whether they play any anti-tumoral role is not determined.

Besides the technical issue discussed below, the TCR repertoire is just analyzed with regards to evenness and richness without much details on the actual TCR sequences. Moreover, since CD4 depletion induces a large expansion of the endogenous CD8 T cells, higher clonal size and decreased diversity are expected. One way to obviate this issue would have been to perform the TCR repertoire analysis on the adoptive transfer system used in fig. 2A to normalize the size of the initial endogenous T cell inoculum.

With regards to the mechanisms of IL-18-Ra high CD8 T cells anti-tumor activity, no attempt is made to study the role of IL-18 triggering using the available anti-IL-18 blocking antibody. How these cells would act is not studied. Like any effector cells they display several effector activities. Which one is involved is not deciphered.

Others:

- The number of cells included in each sample for the repertoire analysis should be provided as repertoire analysis is very dependent upon the number of cells studied. It is not the same to use 100 ng corresponding to the whole cell sample and 100 ng from 1 ug (10 fold more cells). This

point is crucial. The number of reads used for the repertoire analysis should be provided to make sure that it is not limiting in some samples. A complete table of the TCR sequences should be provided for both 3H,I and 5B, C.

- Page 5 line 99: what means "this contrasted with". During the so-called homeostatic expansion, effector differentiation also occurs. In addition, in the current experimental model, IL-2 is provided.

- Page 6 line 133: CD4 depletion increased proliferation but for priming this is not clear

- Page 6 line 135, CD4 depletion barely increased the effector activity of the endogenous CD8 T cells as acknowledged by the authors on page 7 line 154. Panel 3G is not demonstrative as only one data point (with technical replicates) is shown. A cytotoxic response should be analyzed by the shift along the X axis of the E:T ratio.

- In Fig. 5A, one is surprised not see any of the Thy1.1 pos cells in the middle panel.

- The discussion is just a series of wild speculations. For instance:

o A role for DCs on line 224 is irrelevant for this paper

o Unfortunately, a role for Treg is not addressed in this paper (see above)

o The cytokine sink hypothesis is not addressed in this paper

o This study does not show that "cytokine and lymphopenic space resulted in different consequences...". This is not addressed in this paper.

o A role for IL-18 or IL-12 is not addressed in this paper and many other hypothesis could be put forwarded. Even for MAIT or NKT cells, the role of IL-12 and IL-18 in vivo is still a matter of debate.

Reviewer #2:

Remarks to the Author:

The article by Kim et al. present an interesting observation whereby the depletion of CD4+ T cells following adoptive transfer of tumor antigen specific CD8+ T cells leads to enhanced anti-tumor immunity. This is linked to the emergence of an endogenous CD8+ population which is characterised by innate-like characteristics including the expression of IL-18R.

The study is well designed and conclusions are largely well supported. This reviewer makes the following suggestions to further improve the quality of the manuscript.

Major Points

1) Function- The role of the IL-18R+ cells in the capacity of endogenous T cells to control tumor growth should be confirmed directly. In Fig5F, depletion of IL18R+ cells is inferred to reduce the therapeutic effect but this cannot be confirmed without the relevant controls of IL18R+ cells or at least the endogenous T cells control in the same experiment. Is it possible that both IL18R+ and IL18R- subsets are required?

2) Mechanism- Are the IL-18R+ cells inducing anti-tumor immunity in an antigen specific/independent manner. As the authors elude to, they could represent a bystander activated population that targets tumors through antigen independent mechanisms. This could be tested with simple in vitro coculture experiments with antigen negative and positive tumors relative to pmel cells.

3) Cell types- The authors claim that the IL-18R+ cells include NKT cells and MAIT cells, however evidence for this is limited. For MAIT cells this is based upon the frequency of TRAV1/TRAJ33 in the IL-18R enriched cells. Does this increase in a whole population level following CD4 depletion. The data for the NKT TCR is not shown. In both cases the ideal situation would be to use the relevant tetramers to confirm these cells are MAITs/ NK T cells but for NKT cells the proportion of cells expressing NK1.1+TCRbeta+ phenotype should be shown as a minimum. This data should also be presented so it is possible to directly compare the frequency of these cells in CD4 deplete and replete mice. Can the authors comment on whether this is specific to tumor immunity or an inherent feature of CD4 (Treg) depletion.

Minor points

1) On line 116 the authors refer to ".....in the other model (Fig 2D-F)". Could the authors please clarify this as it appears to be the same model as far as I can tell.

- 2) Figure 3A refers to analysis of spleen and lymph nodes and in the remaining panels in Figure 3 it is difficult to determine which organ is being analyzed. Please add labels to the Figures to facilitate this.
- 3) Line 114 "Notably, the presence of CD4post decreased the naïve subset in en-T cells (Fig 3C)." This is overstated as the differences are very minor and statistical significance not shown. Please amend text to reflect this.
- 4) In Fig 3D the authors observe no increase in TIM3 and KLRG1 in lymphoid CD8+ cells and conclude that this indicates that endogenous T cells were not in the late stages of exhaustion. However, this should be clarified/ put into context as these markers of exhaustion/ terminal differentiation are not commonly observed in tumor draining lymph nodes, only the tumors themselves.
- 5) Figure 4D- Please provide greater clarity on when gene expression analysis was performed- was this prior the second transplant as depicted in panel A?
- 6) Fig 1A- Do the authors have data on the subtypes (CD8/ CD4/ Treg) following cyclophosphamide. If so, this would be nice data to include for the context of CD4 depletion.

Reviewer #3:

Remarks to the Author:

Paper by Kim et al. describes the adjuvant effect of depletion of CD4+ T cells in vivo following adoptive transfer of gp100-specific Pmel1 cells in a B16 melanoma model. The authors argue that depletion of Th cells promotes priming and emergence of anti-tumor innate-like IL-18Rhigh cells, mucosal-associated invariant T cells, and bystander-activated T cells. Overall, this study offers additional mechanistic insight into the effects of deeper lymphodepletion on the efficacy of adoptively transferred tumor-specific T cells. The role of competing immune populations on the outcomes of ACT has been described previously and several animal and clinical studies suggest that deeper lymphodepleting conditioning might increase proliferation, persistence and anti-tumor effect of T cells. The major critique of this study is that the authors do not dissect the role of CD4+ T cells in vivo into a more granular picture. This has been done in the past in the Pmel-1 system that the authors use here – with clear demonstration of both possible enhancement and possible impairment of antitumor activity, respectively depending on the presence of only conventional Th cells or presence of T regs (see Antony et. al). This issue is not addressed in this work. It is quite possible that targeting another compartment of the hematopoietic system (e.g. NK, B cells or myeloid cells) would have a similar lymphodepleting effect with enhanced functionality of the transferred CD8+ T cells. Furthermore, while the final experiment of transferring endogenous IL-18RhiCD8+ T cells uses RAG1k/o animals as recipients, it misses the control groups.

- What would the effect of Pmel-1 be in Rag1 ko mice, would CD4+ T cells inhibit or enhance the efficacy?
- Which particular subset of CD4+ T cells would act as negative regulator?
- Would the same effect be observed in CD4 deficient animals treated with pmel-1?
- Would depletion of endogenous CD8+ T cell population impair the efficacy of transferred pmel-1 cells or perhaps it would enhance their anti-tumor effect, as they might also act as a cytokine sink?
- Are the same findings applicable to any other tumor model e.g. MC38 that is known to be inherently more immunogenic?

Detailed minor critique:

CD4post enhances ACT efficacy

Fig.1 Anti-CD4 post-conditioning increases anti-melanoma responses in adoptive T cell therapy

\ Ex-vivo priming of CD8+ T cells for only 2 days? This is not the standard protocol for pmel-1 experiments for many other researchers.

\ Please provide numbers of CD8+ T cells transferred.

\ Fig.1A: no error bars in the spleen graph, and lymph node 300mg/kg condition, after 10 days. Does it mean only 1 mouse was analyzed?

\ Fig.1C: shows very slow tumor growth (around day 15 in CTRL, day 20 for other conditions), maybe due to the low tumor initial number of B16 cells injected. This looks more like a protection than therapy experiments. Please provide a rationale for the quantity of tumor cells injected.

\ Furthermore, no CTX-only condition is shown, to compare with CTX/CD4.

\ In group CTX/ex-T/CD4, the two tumor-free mice seem not to have had tumor at all. Was the

tumor presence asserted in any way?

\ Please show tumor area (mm²) rather than volume (mm³): this is the more standard and less biased way of measuring the responses in pmel-1 model.

Fig. 2. Anti-CD4 post-conditioning accelerates the division and intratumoral infiltration of ex vivo primed and endogenous CD8+ T cells.

\ Please comment on selecting day 25 as endpoint.

\ Please provide FACS gates for CD44^{hi}CD62L^{lo} cells (in supp. Fig. this specific population is not shown).

\ Correct legend in Supp.Fig.2B, to be consistent with legend in Fig.2C

\ Please comment on the higher tumor cell dose used in Fig.2G-J

\ Fig.2G: Please provide numbers of CD8+ T cells injected.

\ Fig.2J: please change colors, or line shape, of histograms, to make it more distinguishable.

Fig. 3. Anti-CD4 post-conditioning increases the differentiation and tumor-reactivity of endogenous CD8+ T cells.

\ Please provide FACS gates (dot plots) for populations analyzed: 4-1BB, CD69, CD44^{hi}CD62L^{lo}

\ Supp.Fig.3A/C: FACS gate for IL-2 is not convincing – there is very little signal. Please provide the negative control used to set it up (as well as for other cytokines). Please also provide gate strategies for Supp.Fig.3B/D.

\ Fig.3E: please choose a different color scheme that could be easier to read for all readers (the reviewer is colorblind).

\ Fig.3D: same comment as Fig.2J

\ Fig.3H: the message here is difficult to get at first glance, maybe the authors might want to use a different type of graph.

Fig. 4. Anti-CD4 post-conditioning enriches IL-18R^{hi} endogenous CD8+ T cells with potent effector potential.

\ Please provide numbers of CD8+ T cells/En-T cells injected.

\ Fig.4B: did the 2 tumor-free mice actually show any growth of tumor cells?

\ Fig.4G: please correct legend, are IL-18R^{lo} and IL-18R^{hi} respectively IL-18R^{mid} and IL-18R^{high}?

\ Fig.4G: See comments Fig.3E and Fig.3A/C.

Fig. 5. IL-18R^{hi} endogenous CD8+ T cell subset with innate-like property has potent anti-tumor effect.

\ Please provide numbers of CD8+ T cells injected.

\ In the flow through, 20% of the cells are IL-18R^{high}. Was this contamination addressed while analyzing TCR and gene expression?

\ Where iNKT and MAIT characterized by their phenotype, in addition to TCR clonotypes and gene enrichment?

Discussion:

The activated en-T cells that express high level of IL-18R include iNKT, MAIT, NKG2D+ T cells, and TVM, which show cytokine-mediated anti-tumor responses.

\ There is no cytokine-mediated anti-tumor response, related to these cells, shown.

General Comments:

\ Were other models tested? And why, with the pmel mouse model, no vaccination was used?

Methods:

Cells

\ Please state viability and confluence of B16 cells at the moment of injection.

Using CD8 microbeads (positive selection beads; Miltenyi Biotech, CA, USA), CD8+ T cells were isolated from lymphocytes. For ex vivo priming, Thy1.1+ Pmel-1 CD8+ T cells were further stimulated with 5 µg/mL of human gp10025–33 peptide (KVPRNQDWL; Peptron, Daejeon, Korea) in the presence of 5% splenocytes for 2 days.

\ Were the CD8+ T cells isolated only from lymphocytes or, as stated in the line before, also from spleen? Is the 2 days enough to get a proper priming of CD8+ T cells? Were longer conditions tested? And which medium was used for the priming? Finally, were the splenocytes irradiated?

\ Why such a short stimulation?

Adoptive transfer models

On day 5, mice received intravenous injections of CD8+ T cells.

\ Please provide number of injected cells, and localization of the injection.

\ Please add a paragraph related to Fig.4A

Flow cytometry

The isolated cells were placed in the well (5×10^5 /well) of a 96-well round-bottom plate, treated with a Cell Stimulation Cocktail and Protein Transport Inhibitor Cocktail (Thermo Fisher Scientific, MA, USA), and then incubated for 16 h.

\ This is not a typical length of in vitro stimulation with this cocktail (see package insert). Overall PMI/ionomycin and protein transport inhibitors are toxic to T cells if left in culture for too much time, likely biasing the results.

We thank the reviewers for their time and for providing valuable suggestions. To address the key concerns raised by the reviewers, we have dissected the variables involved in our model and evaluated the contribution of each variable to the generation of the IL-18R α^{hi} subset. The antigen specificity and direct anti-tumor effect of IL-18R α^{hi} cells have also been investigated using additional *in vivo* experiments. In this process, we have found interesting results that provide new insights regarding the true nature of IL-18R α^{hi} CD8 $^{+}$ T cells. We have substantially revised the original manuscript and reorganized the figures based on these findings. Please find the detailed point-by-point responses to the reviewers' comments as follows.

Reviewer #1 (Remarks to the Author): with expertise in T cells and innate T cells

Reviewer comment

This paper develops an experimental model of tumor rejection using an adoptive transfer of ex-vivo activated tumor specific CD8 T cells associated with pre-transfer lympho-depletion followed by IL-2 treatment and CD4 depletion. CD4 depletion on the top of lympho-depletion leads to a better anti-tumor effect which is attributed to the expansion of "innate-like" CD8 T cells. These IL-18Ra high CD8 T cells are characterized at the phenotypic and transcriptional level. These IL-18-Ra high CD8 T cells encompasses NKT, MAIT and "bystander" memory T cells.

Although the model is complex and artificial, on a whole, the work is interesting but several conclusions are not fully supported by the data. The phenotypic, lymphokine expression and transcriptome characterization of the different subsets is correctly performed but does not provide real new insight besides the fact that cells that have extensively proliferated display effector or memory features with the corresponding phenotypic and functional properties.

Author response

We appreciate the comments. As the reviewer mentions, the original manuscript provided limited information regarding the IL-18R α^{hi} cells, as it presented data only regarding the exhibition of enriched innate-like markers/TCRs. We have now investigated the role of TCR/IL-18 signaling and other variables in the function/generation of IL-18R α^{hi} CD8 $^{+}$ T cells. To the best of our abilities, here, we have provided data from additional experiments and have responded to the concerns and suggestions raised by the reviewer.

Reviewer comment

Most experiments seem to have been performed only once (with 4-5 data points each) and the number of mice is often very limited. This is a very serious issue. Science is about reproducibility and every experiment should be performed at least twice. This applies also to the mouse experiments. Two groups in fig. 1C included 10 mice while the number of mice is much lower in the other experiments. It is nowhere stated that the experiment was reproducible (5 mice twice is better than 10 mice once). The key experiment displayed in 5F is not only performed only once but also does not contain the

positive control performed in 4B. This is incorrect. Comparison between experiments is deduction but not a demonstration.

Author response

Per the reviewer's comment, there were concerns regarding the reliability of the *in vivo* anti-tumor efficacy of the CTX^{pre}/CD4^{post}-induced IL-18R α ^{hi} subset. Hence, in the revised manuscript, we have attempted to clearly show that the key results are genuine and reproducible. First, we conducted an additional experiment for confirming the key result of this study (i.e., the evaluation of CTX^{pre}/ex-T/CD4^{post} combination) with a larger number of animals (Fig. 1c-f; reproducible in three independent experiments). Next, to verify the role of the key component (IL-18R α ^{hi} cells), we evaluated the direct activity of the IL-18R α ^{hi}-enriched subset instead of using the IL-18R α ^{hi}-depleted cell population, as was done in the original manuscript (Fig. 5c, h and supplementary Figs. 7 and 8; reproducible in two independent experiments). Finally, the role of the CTX^{pre}/CD4^{post}-induced cytokine milieu that continuously produces short-term IL-18R α ^{hi} cells was examined by disrupting IL-18 signaling (anti-IL-18/IL18r1 KO mice), which was crucial in augmenting the function of the IL-18R α ^{hi} CD8⁺ T cells (Fig. 5j, k; reproducible in two independent experiments). We have added these data to the figures of the revised manuscript.

Reviewer comment

It is not clear whether the experimental model is representative of the human situation in which fludarabine is often used and leads to a long-term CD4 depletion.

Author response

As pointed out by the reviewer, fludarabine (along with cyclophosphamide) is frequently used in the human setting. However, the lymphodepletion effect is not significant in mouse models (see attached figure from Johnson et al. [PMID: 29636345]). Nevertheless, considering the strong correlation between IL-18R α ^{hi} ex-T ratio and lymphopenia score (Fig. 6c), any type of systemic lymphodepletion may possibly result in considerable synergy with CD4^{post}/ACT.

The extent of lymphodepletion induced by various agents. C57BL6 mice were treated with the indicated agents. The total number of host splenocytes were measured 4 days after treatment (left) or longitudinally (right). Dose of each treatment: fludarabine (FLU), 4 mg/mouse; cyclophosphamide

(CTX), 300 mg/kg; total body irradiation (TBI), 9 Gy.

Reviewer comment

The model is complex with at least 6 variables

- *In vitro primed tumor reactive CD8 T cells*
- *Lympho depletion*
- *IL-2*
- *CD4 depletion (decreasing both Tregs and conventional CD4)*
- *Endogenous conventional CD8 T cells*
- *Endogenous IL-18-Ra high CD8 T cells*

*How these variables interact with each other is unclear and the authors focus mainly on the CD4 T cell depletion but other parameters may be involved. How CD4 depletion leads to the inflation of IL-18-Ra high CD8 T cells is not deciphered. Is injection of IL-2 necessary? This can easily be tested by doing the experiment without IL-2 injection. Is Treg depletion involved? This can be tested by using *Foxp3-DTR* mice. Is there any role for the tumor specific T cells? This can be tested by not doing this transfer.*

Author response

We appreciate the valuable suggestion. We attempted to identify the variables that significantly affect the generation of IL-18R α^{hi} cells (which elicited IL-18 signaling-dependent anti-tumor efficacy in Fig. 5f, h, j, k). In brief, five variables (the presence of tumor, CTX^{pre}, ex-T, IL-2, and CD4^{post}) were evaluated by subtracting each variable from the full regimen (see attached table below). Additionally, the modulation of IL-18 signaling in whole cells (anti-IL-18) or endogenous cells (*Il18r1* KO mice), Treg-specific deletion (*Foxp3^{DTR}*), and permanent lymphopenic condition (*Rag1* KO mice) were tested to identify the key factors involved in IL-18R α^{hi} T cell generation. The result indicated that CTX^{pre}, IL-2 injection, and CD4^{post} significantly affected ex-T cells with respect to the frequency and absolute number of IL-18R α^{hi} cells (Fig. 6a, b). Intriguingly, *Foxp3^{DTR}* mice also enriched IL-18R α^{hi} ex-T cells that was comparable with that observed with the full regimen, implying the role of Treg depletion in IL-18R α^{hi} generation. Furthermore, the extent of lymphopenia correlated strongly with the proportion of IL-18R α^{hi} ex-T cells (Fig. 6c). Based on these observations, we have concluded that both the absence of Treg and lymphopenia play an important role in generating sufficient amounts of IL-18R α^{hi} ex-T cells. We have added these data to the figure and revised the manuscript (page 10, line 16 to page 11, line 8).

Recipient	WT	WT	WT	WT	WT	WT	WT	Il18r1 KO	Foxp3^{DTR}	Rag1 KO
Tumor	+	+	+	+	+	-	+	+	+	+
CTX ^{pre}	+	+	+	+	-	+	+	+	+	+
Ex-T	+	+	+	-	+	+	+	+	+	+
IL-2	+	+	-	+	+	+	+	+	+	+
CD4 ^{post}	+	-	+	+	+	+	+	-	-	-
other							α IL-18		DT	

Reviewer comment

Another concern is about the antigen specificity of the IL-18-Ra high CD8 T cells and the mechanisms of their involvement in tumor rejection. The authors observe an important inflation of IL-18-Ra high CD8 T cells which appear to be an heterogeneous subset with regards to antigen specificity. This is not really deciphered (no CD1d/ α GalCer or MR1/5-OP-RU tetramer staining) and no use of the available CD1d or MR1 KO mice. The so-called bystander CD8 T cells are not analyzed further. Whether they play any anti-tumoral role is not determined.

Author response

We have conducted *in vivo* efficacy test of IL-18R α^{hi} en-T cells in the presence of anti-MHC I treatment to determine the involvement of TCR in the anti-tumor activity. The anti-tumor effect of IL-18R α^{hi} en-T cells disappeared in the presence of MHC I inhibitor, verifying that the cells suppress tumor via TCR–MHC I interaction (Fig. 5c). TCR dependency of the IL-18R α^{hi} subset function was further corroborated by the *in vitro* experiment, in which IL-18R α^{hi} ex-T cells showed peptide-specific strong effector functions (Fig. 5f). Overall, IL-18R α^{hi} en-T cells possibly induced tumor rejection via the interaction between TCR and melanoma-antigen loaded on MHC I. We have added these data to the figure and revised the manuscript (page 8, line 18 to page 9, line 11).

Considering the importance of TCR–MHC I interaction, it is interesting to note that TCR clonotypes related to iNKT and MAIT were enriched in the IL-18R α^{hi} subset (Supplementary Fig. 6). This change in the TCR repertoire was confirmed by the result of CD1d/ α GalCer staining (see attached figure below). Considering that IL-18R α^{hi} cells included heterogeneous subsets that could be expanded by TCR-independent cytokine signaling (e.g., iNKT, MAIT, bystander-activated, and T_{VM}-like phenotype), we speculated that this is due to the proliferation bias of these subsets in the CTX^{pre}/CD4^{post}-induced milieu. We have added the discussion on this issue in the manuscript (page 13, line 26 to page 14, line 1).

Binding of CD1d/ α GalCer tetramer to En-T cells. En-T cells from CTX^{pre}- and CTX^{pre}/CD4^{post}-experienced mice were stained with CD1d/ α GalCer tetramer:PE and anti-IL-18R α :APC. (Left) Representative staining data. (Right) Summary of total staining data (n=5 mice per group). **, p<0.01; ***, p<0.001; two-tailed unpaired Student's t-test.

Reviewer comment

Besides the technical issue discussed below, the TCR repertoire is just analyzed with regards to

evenness and richness without much details on the actual TCR sequences. Moreover, since CD4 depletion induces a large expansion of the endogenous CD8 T cells, higher clonal size and decreased diversity are expected. One way to obviate this issue would have been to perform the TCR repertoire analysis on the adoptive transfer system used in fig. 2A to normalize the size of the initial endogenous T cell inoculum.

Author response

Thank you for the suggestion. Initially (in the original manuscript), we conducted TCR analysis to study the nature of IL-18R α ^{hi} en-T cells, as they appeared to possess innate-like properties and play an important role in anti-melanoma response. However, during the revision, we found that IL-18R α ^{hi} ex-T cells were the main contributors of CTX^{pre}/CD4^{post}-induced anti-tumor efficacy. IL-18 inhibition in CTX^{pre}/CD4^{post}-exposed wild-type mice (blocks activity of both IL-18R α ^{hi} en-T/ex-T cells) did not have any therapeutic benefit, whereas CTX^{pre}/CD4^{post}-exposed *Il18r1* KO mice (only blocks the activity of IL-18R α ^{hi} en-T cells) efficiently suppressed tumor growth (Fig. 5j, k). Based on this and other related results, we have decided to de-emphasize the significance of the TCR analysis, as well as the role of iNKT/MAIT in the revised manuscript, as these appear to be less significant bystander effects of the CTX^{pre}/CD4^{post}-induced milieu. Thus, we have removed all the data except that related to the comparison of Shannon evenness, which reflects CTX^{pre}/CD4^{post}-induced expansion (Figs. 3e and 4e). Accordingly, we have revised figures and the manuscript (page 6, line 18 to 23 and page 7, line 20 to 23).

Reviewer comment

With regards to the mechanisms of IL-18-Ra high CD8 T cells anti-tumor activity, no attempt is made to study the role of IL-18 triggering using the available anti-IL-18 blocking antibody. How these cells would act is not studied. Like any effector cells they display several effector activities. Which one is involved is not deciphered.

Author response

As the reviewer points out, the study of IL-18 signaling is possibly an important factor in IL-18R-expressing T cell functions. Hence, we used several approaches to study the effect of IL-18 signaling: IL-18-blocking antibody (inhibits IL-18 signal in both ex-T/en-T cells), *Il18r1* KO mice (inhibits IL-18 signal in en-T cells), and recombinant IL-18 (direct IL-18 signal induction). The effect of IL-18 signaling on IL-18R α ^{hi} cell functions was observed in the *in vitro* assay, in which the addition of recombinant IL-18 increased the secretion of several effector cytokines (Fig. 5f). Evidences from two *in vivo* experiments using IL-18-blocking antibody and *Il18r1* KO mice further confirmed this effect. First, the anti-tumor effect induced by the transfer of IL-18R α ^{hi} ex-T cells was reduced in anti-IL-18-treated mice (Fig. 5h). Next, the key regimen in the current study, CTX^{pre}/ex-T/CD4^{post}, lost most of its therapeutic properties when it accompanied anti-IL-18 treatment (Fig. 5j, k). Notably, these inhibitions were not observed in *Il18r1* KO mice, implying the more crucial role of IL-18R α ^{hi} ex-T cells than IL-18R α ^{hi} en-T cells. Collectively, we have concluded that the IL-18R α ^{hi} subset drives strong tumor-suppressive activity in IL-18 signaling-dependent manner. We have added these data to the figures and revised the manuscript (page 9, line 8 to page 10, line 14).

Reviewer comment

- The number of cells included in each sample for the repertoire analysis should be provided as repertoire analysis is very dependent upon the number of cells studied. It is not the same to use 100 ng corresponding to the whole cell sample and 100 ng from 1 ug (10 fold more cells). This point is crucial. The number of reads used for the repertoire analysis should be provided to make sure that it is not limiting in some samples. A complete table of the TCR sequences should be provided for both 3H,I and 5B, C.

Author response

We appreciate the detailed advice for the analysis of the TCR repertoire. In the original manuscript, we had investigated the TCR repertoire of en-T cells to identify the nature of IL-18R α^{hi} en-T cells, which showed innate-like properties. Additional experiments during the revision revealed that TCR–MHC I interaction is crucial for the function of these cells (Fig. 5c). In addition, IL-18R α^{hi} ex-T cells, which have TCR of defined specificity (gp100), were more important than en-T cells in anti-B16-F10 response (Fig. 5j, k; comparison between CTX $^{\text{pre}}$ /CD4 $^{\text{post}}$ -experienced WT, *I18r1* KO, and WT+anti-IL-18). Consequently, the significance of TCR analysis of en-T cells has diminished, as it does not provide any new information other than confirming repertoire skewing by anti-CD4, which has been reported in previous studies (PMID: 30733724). Therefore, we have retained only the results pertaining to Shannon evenness (Figs. 3e and 4e; page 6, line 18 to 23 and page 7, line 20 to 23). In addition, to resolve the concern regarding TCR reads and cell population size, we have recalculated Shannon evenness from down-sampled data (to 10,000 reads). The information regarding cell numbers, input RNAs, and raw TCR reads have been added to the revised manuscript (page 20, line 12 to 26).

Reviewer comment

- Page 5 line 99: what means "this contrasted with". During the so-called homeostatic expansion, effector differentiation also occurs. In addition, in the current experimental model, IL-2 is provided.

Author response

As pointed out by the reviewer and as suggested by Muranski et al. (PMID: 17139318), naive T cells change to CD44 $^+$ CD62L $^-$ phenotype when exposed to homeostatic cytokines, self-peptides, and other antigens in an immunodepleted environment. Hence, we have removed the description from the manuscript.

Reviewer comment

- Page 6 line 133: CD4 depletion increased proliferation but for priming this is not clear

Author response

We agree with the reviewer that whether CD4 $^{\text{post}}$ elevated priming is not clear, although en-T cells exhibited several indirect evidences of priming, such as the upregulation of activation/exhaustion

markers and reduction in naïve phenotype. Hence, we have removed the term “priming” to clear the confusion.

Reviewer comment

- Page 6 line 135, CD4 depletion barely increased the effector activity of the endogenous CD8 T cells as acknowledged by the authors on page 7 line 154. Panel 3G is not demonstrative as only one data point (with technical replicates) is shown. A cytotoxic response should be analyzed by the shift along the X axis of the E:T ratio.

Author response

Adequate statistical analysis has been conducted using two-way ANOVA analysis and the Bonferroni post-hoc test. The detailed information has been added to the revised manuscript (Fig. 3d; page 33, line 8 to page 34, line 1).

Reviewer comment

- In Fig. 5A, one is surprised not see any of the Thy1.1 pos cells in the middle panel.

Author response

We have corrected the figure (Fig. 4d in the revised manuscript).

Reviewer comment

*- The discussion is just a series of wild speculations. For instance:
o A role for DCs on line 224 is irrelevant for this paper
o Unfortunately, a role for Treg is not addressed in this paper (see above)
o The cytokine sink hypothesis is not addressed in this paper
o This study does not show that "cytokine and lymphopenic space resulted in different consequences...". This is not addressed in this paper.
o A role for IL-18 or IL-12 is not addressed in this paper and many other hypothesis could be put forwarded. Even for MAIT or NKT cells, the role of IL-12 and IL-18 in vivo is still a matter of debate.*

Author response

The manuscript has been substantially revised after incorporating the new findings from the additional experiments. Accordingly, we have also revised the discussion to include the description addressed in this study and have removed irrelevant sentences.

- o The role for DCs has been removed.
- o We have discussed the effect of Treg in this study by describing the data obtained from *Foxp3*^{DTR} mice (page 12, line 25 to page 13, line 7).
- o As there is no direct evidence of cytokine sink-related effect, we have only described a hypothetical role of the cytokine sink in the current study by citing references (page 13, line 18 to page 14, line 8)

- o The discussion on the effect of lymphopenic space has been removed.
- o We have discussed the role of IL-18 on the function of the IL-18R α^{hi} subset (page 13, line 8 to 17).

Reviewer #2 (Remarks to the Author): with expertise in T cell adoptive transfer – immunotherapy

Reviewer comment

The article by Kim et al. present an interesting observation whereby the depletion of CD4⁺ T cells following adoptive transfer of tumor antigen specific CD8⁺ T cells leads to enhanced anti-tumor immunity. This is linked to the emergence of an endogenous CD8⁺ population which is characterised by innate-like characteristics including the expression of IL-18R.

The study is well designed and conclusions are largely well supported. This reviewer makes the following suggestions to further improve the quality of the manuscript.

Author response

We appreciate the valuable suggestions made by the reviewer. Overall, we have conducted additional experiments and revised the manuscript to elucidate the role, mechanism, and identity of the IL-18R^{hi} cell subset. Please find the following point-by-point responses to the reviewer's comments, which describe the new findings derived from the additional experiments and indicate the revised parts in the manuscript.

Reviewer comment (major points)

1) Function- The role of the IL-18R⁺ cells in the capacity of endogenous T cells to control tumor growth should be confirmed directly. In Fig5F, depletion of IL18R⁺ cells is inferred to reduce the therapeutic effect but this cannot be confirmed without the relevant controls of IL18R⁺ cells or at least the endogenous T cells control in the same experiment. Is it possible that both IL18R⁺ and IL18R⁻ subsets are required?

2) Mechanism- Are the IL-18R⁺ cells inducing anti-tumor immunity in an antigen specific/independent manner. As the authors elude to, they could represent a bystander activated population that targets tumors through antigen independent mechanisms. This could be tested with simple in vitro coculture experiments with antigen negative and positive tumors relative to pmel cells.

Author response

We have attempted to clarify the anti-tumor role and the mechanism of action of IL-18R^{hi} endogenous CD8⁺ T cells in additional experiments. To directly evaluate the anti-tumor role, IL-18R^{hi} en-T cells were sorted and transferred to tumor-bearing *Rag1* KO mice. A group was treated with the cells in the presence of anti-MHC I antibody to check TCR dependency of the anti-tumor function. The result showed the TCR-dependency of IL-18R^{hi} en-T cells in tumor suppression (Fig. 5c and Supplementary Fig. 7). This was further confirmed by other experiments, in which IL-18R^{hi} ex-T cells (which have gp100-specific TCR) exhibited TCR/IL-18 signaling-dependent anti-tumor effector function *in vitro* and *in vivo* (Fig. 5f, h and Supplementary Fig. 8). Additionally, IL-18R^{hi} ex-T cells were more important than IL-18R^{hi} en-T cells, as CTX^{pre}/ex-T/CD4^{post}-experienced *Il18r1*

KO mice (which have no IL-18R α^{hi} en-T cells) exhibited robust tumor suppression (Fig. 5j, k). In summary, we have found that the IL-18R α^{hi} CD8 $^{+}$ T cell subset shows TCR/IL-18 signal-dependent anti-tumor activity in melanoma. We have added these data to the figure and revised the manuscript (page 8, line 18 to page 10, line 14).

Reviewer comment (major points)

3) Cell types- The authors claim that the IL-18R $^{+}$ cells include NKT cells and MAIT cells, however evidence for this is limited. For MAIT cells this is based upon the frequency of TRAV1/TRAJ33 in the IL-18R enriched cells. Does this increase in a whole population level following CD4 depletion. The data for the NKT TCR is not shown. In both cases the ideal situation would be to use the relevant tetramers to confirm these cells are MAITs/ NK T cells but for NKT cells the proportion of cells expressing NK1.1+TCRbeta+ phenotype should be shown as a minimum. This data should also be presented so it is possible to directly compare the frequency of these cells in CD4 deplete and replete mice. Can the authors comment on whether this is specific to tumor immunity or an inherent feature of CD4 (Treg) depletion.

Author response

As suggested by the reviewer, we performed CD1d/ α GalCer tetramer staining of CTX $^{\text{pre}}$ /CD4 $^{\text{post}}$ -experienced T cells and confirmed the expression of IL-18R α on iNKTs (see attached figure below). As our proposed mechanism suggests TCR/IL-18 signal-dependent anti-tumor response of IL-18R α^{hi} CD8 $^{+}$ T cells (as in the author responses to the previous comments), we anticipated that IL-18R-upregulation in MAITs/iNKTs (which have TCRs specific for unrelated antigens) is possibly an inherent feature induced by the CTX $^{\text{pre}}$ /CD4 $^{\text{post}}$ -induced milieu. More specifically, MAITs/iNKTs appear to upregulate IL-18R via increased availability of the IL-7 signal as shown in Fig. 6h, as these cells express high levels of IL-7 receptor (PMID: 18354162 and 23447689). We have revised the manuscript to include the discussion on this issue (page 13, line 26 to page 14, line 1).

Binding of CD1d/ α GalCer tetramer to En-T cells. En-T cells from CTX $^{\text{pre}}$ - and CTX $^{\text{pre}}$ /CD4 $^{\text{post}}$ -experienced mice were stained with CD1d/ α GalCer tetramer:PE and anti-IL-18R α :APC. (Left) Representative staining data. (Right) Summary of total staining data (n=5 mice per group). **, p<0.01; ***, p<0.001; two-tailed unpaired Student's t-test.

Reviewer comment (minor points)

1) On line 116 the authors refer to “.....in the other model (Fig 2D-F)”. Could the authors please clarify this as it appears to be the same model as far as I can tell.

Author response

The following schematic diagrams show the differences between the two models. Compared with the left model (Fig. 2a-d in the revised manuscript), the right model (Fig. 2e-j in the revised manuscript) contains an additional cell transfer (i.e., fluorescence-labeled polyclonal CD45.1⁺ CD8⁺ T cells). This step was added to trace the proliferation rate and the resulting phenotypical change of polyclonal CD8⁺ T cells. The manuscript has been revised to clearly describe the purposes and differences between the two models (page 5, line 10 to 15).

Schematic of the experiment conducted in this study. (Left) C57BL/6 mice inoculated with 2×10^5 B16-F10 melanoma were injected with 2×10^6 *ex-vivo* primed Thy1.1⁺ Pmel-1 CD8⁺ T (ex-T) cells and IL-2 (10,000 IU). Two days prior to ex-T cell transfer, mice were pre-conditioned with 300 mg/kg CTX. CD4^{post} group received weekly treatment of anti-CD4 antibody from day 10. (Right) C57BL/6 mice inoculated with 2×10^5 B16-F10 melanoma were infused with the mixture of 1×10^6 CFSE-labeled *ex-vivo* primed Thy1.1⁺ Pmel-1 CD8⁺ T cells (Thy1.1⁺) and 1×10^6 Celltrace Violet-labeled CD45.1⁺ CD8⁺ T cells (CD45.1⁺). IL-2 (10,000 IU) was daily injected from day 5 to day 7. Two days before the ex-T cell transfer, mice were pre-conditioned with 300 mg/kg CTX. The CD4^{post} group received weekly treatment of anti-CD4 antibody from day 10.

Reviewer comment (minor points)

2) Figure 3A refers to analysis of spleen and lymph nodes and in the remaining panels in Figure 3 it is difficult to determine which organ is being analyzed. Please add labels to the Figures to facilitate this.

Author response

We have analyzed the mixed cell population of spleen and inguinal lymph nodes if not otherwise specified. To avoid confusion, we have added this information in the manuscript (page 16, line 15 to 16).

Reviewer comment (minor points)

3) Line 114 “Notably, the presence of CD4^{post} decreased the naïve subset in en-T cells (Fig 3C).” This is overstated as the differences are very minor and statistical significance not shown. Please amend text to reflect this.

Author response

We apologize for the confusion caused by the figure. It was difficult to distinguish the memory phenotype of ex-T (not significant) and en-T cells (significant). We have changed the position of the figure (Fig. 2c).

Reviewer comment (minor points)

4) In Fig 3D the authors observe no increase in TIM3 and KLRG1 in lymphoid CD8⁺ cells and conclude that this indicates that endogenous T cells were not in the late stages of exhaustion. However, this should be clarified/ put into context as these markers of exhaustion/ terminal differentiation are not commonly observed in tumor draining lymph nodes, only the tumors themselves.

Author response

As the reviewer suggested, we have analyzed the exhaustion markers in tumor-infiltrating CD8⁺ T cells (see attached figure below). The result showed that most ex-T and en-T cells expressed PD-1, and that the differences in exhaustion marker expression between the CTX^{pre} and CTX^{pre}/CD4^{post} groups were lower than that in lymphoid tissues (Fig. 2d). Therefore, we concluded that the upregulated exhaustion markers in lymphoid tissues represent the different activation status (e.g., hyperactivation) rather than exhaustion. We have revised the manuscript based on this observation (page 5, line 4 to 9).

The expression of exhaustion markers in CD8⁺ T cells infiltrated in tumor tissue. Ex-T and en-T cells in tumor tissues from CTX^{pre}- and CTX^{pre}/CD4^{post}-experienced mice were analyzed as in Fig. 3f-i.

Reviewer comment (minor points)

5) Figure 4D- Please provide greater clarity on when gene expression analysis was performed- was this prior the second transplant as depicted in panel A?

Author response

We apologize for the confusion regarding the experimental method. We have revised the manuscript to clearly present the information (Fig. 4d; page 7, line 20 to 21).

Reviewer comment (minor points)

6) Fig 1A- Do the authors have data on the subtypes (CD8/ CD4/ Treg) following cyclophosphamide. If so, this would be nice data to include for the context of CD4 depletion.

Author response

We have analyzed the kinetics of CD8/CD4/Treg following CTX treatment. All subtypes were initially depleted using high dose (300 mg/kg) of CTX and the recovery curve specific to each subtype was drawn. We have added these data to the figure (Supplementary Fig. 1a, b).

Reviewer #3 (Remarks to the Author): with expertise in T cell adoptive transfer – immunotherapy

Paper by Kim et al. describes the adjuvant effect of depletion of CD4⁺ T cells in vivo following adoptive transfer of gp100-specific Pmel1 cells in a B16 melanoma model. The authors argue that depletion of Th cells promotes priming and emergence of anti-tumor innate-like IL-18Rhi cells, mucosal-associated invariant T cells, and bystander-activated T cells. Overall, this study offers additional mechanistic insight into the effects of deeper lymphodepletion on the efficacy of adoptively transferred tumor-specific T cells. The role of competing immune populations on the outcomes of ACT has been described previously and several animal and clinical studies suggest that deeper lymphodepleting conditioning might increase proliferation, persistence and anti-tumor effect of T cells. The major critique of this study is that the authors do not dissect the role of CD4⁺ T cells in vivo into a more granular picture. This has been done in the past in the Pmel-1 system that the authors use here – with clear demonstration of both possible enhancement and possible impairment of antitumor activity, respectively depending on the presence of only conventional Th cells or presence of T regs (see Antony et. al). This issue is not addressed in this work. It is quite possible that targeting another compartment of the hematopoietic system (e.g. NK, B cells or myeloid cells) would have a similar lymphodepleting effect with enhanced functionality of the transferred CD8⁺ T cells. Furthermore, while the final experiment of transferring endogenous IL-18RhiCD8⁺ T cells uses RAG1k/o animals as recipients, it misses the control groups.

Author response

We appreciate the reviewer's comments and suggestions regarding this study. Throughout the study, our primary goal has been to design a practical immunomodulation regimen that transiently elicits characteristic changes in the transferred cells in ACT. Therefore, rather than optimizing subset composition (e.g., Th, Treg) for overall anti-tumor immunity, we have chosen whole CD4 depletion that instantly affects CD8⁺ T cells in various way. Nevertheless, it will be interesting to study the effect of each subset-specific depletion because, as mentioned by the reviewer, CD4⁺ T cells perform two opposite functions: pro-tumor activity by Tregs and anti-tumor activity by Th subset. Therefore, we have conducted additional experiments using *Foxp3*^{DTR} mice and found that Treg depletion can substitute CD4 depletion in enriching the key component of this study, i.e., IL-18R^{hi} CD8⁺ T cells. Considering this data, we have revised the manuscript to emphasize the effect of Tregs on IL-18R^{hi} CD8⁺ T cells. Here we provide the point-by-point responses to the reviewer's comments and mention the related revisions in the manuscript.

Reviewer comment

- *What would the effect of Pmel-1 be in Rag1 ko mice, would CD4⁺ T cells inhibit or enhance the efficacy?*

Author response

It will be interesting to study the role of the CD4⁺ subset by adding it to immune-deficient mice (such as *Rag1* KO) instead of depleting the subset from immune competent mice, as was shown in the

original manuscript. As CD4 depletion increased ACT efficacy, CD4⁺ T cell transfer with Pmel-1 may decrease the therapeutic effect. Our unpublished data from a similar model showed that the incorporation of polyclonal CD4⁺ T cells reduced the efficacy of the combination of radiation pre-conditioning and Pmel-1 transfer (see the attached figure below). It is noteworthy that the extent of inhibitory effect shown in the current study was reduced probably because the number of CD4⁺ T cells is higher in immune competent mice. This effect is possibly induced by Tregs in the polyclonal CD4⁺ T cells as demonstrated by Anthony et al. (PMID: 15728465). In addition, the mechanism may also involve the reduction in IL-18R α^{hi} Pmel-1 cells, as demonstrated in the present study (Fig. 6a, b). Nonetheless, we anticipated that if the infused CD4⁺ T cells were not polyclonal nor heterogeneous (e.g., tumor-reactive TCRs or Treg-depleted Th cells), the ACT efficacy would increase as reported in other studies (PMID: 15728465 and 24114780).

Anti-tumor effect of antigen-specific T cell and polyclonal CD4⁺ or CD8⁺ T cells. *Rag1* KO mice inoculated with 2×10^5 B16-F10 melanoma were infused with 2×10^6 activated Thy1.1⁺ Pmel-1 CD8⁺ T (Pmel-1) cells. Some mice were additionally infused with 2×10^6 polyclonal naive CD4⁺ or CD8⁺ T cells. Two days before the cell transfer, mice were pre-conditioned with 4 Gy total body irradiation. Mice were daily treated with 10,000 IU IL-2 from day 5 to day 9.

Reviewer comment

- Which particular subset of CD4⁺ T cells would act as negative regulator?

Author response

The present study demonstrates that the tumor-specific IL-18R α^{hi} CD8⁺ T cells are the key components in increasing ACT efficacy (Fig. 5c, h, j, k). Hence, we have investigated the Treg-specific effect on the generation of IL-18R α^{hi} subset using *Foxp*^{DTR} mice. The depletion of Treg and CD4⁺ subset did not show significant difference in the proportion of IL-18R α^{hi} ex-T and en-T cells, implying the crucial role of Tregs in this mechanism. However, Treg-depleted mice showed severe autoimmune symptoms and died before 3 weeks post Treg-depletion (Supplementary Fig. 12). This is possibly due to some Th subsets that lead to autoimmune inflammation, as demonstrated in previous studies (PMID: 19890056 and 33039897). Therefore, we concluded that while Treg is a negative regulator of IL-18R α^{hi} subset generation, whole CD4⁺ T cell depletion should be included in the regimen to have overall clinical benefit. We have added these data to the figure and have revised the manuscript (Supplementary Fig. 12; page 10, line 16 to 27 and page 12, line 25 to page 13, line 7).

Reviewer comment

- Would the same effect be observed in CD4 deficient animals treated with pmel-1?

Author response

As CD4 KO mice are defective in CD4⁺ T cell development, we anticipated that CD4 KO mice might show similar therapeutic efficacy when combined with CTX^{prc} and Pmel-1 transfer. Additionally, an experiment with similar setting showed that continuous treatment with anti-CD4 for the entire experimental period (~80 days) did not compromise the therapeutic effect (data not shown).

Reviewer comment

- Would depletion of endogenous CD8⁺ T cell population impair the efficacy of transferred pmel-1 cells or perhaps it would enhance their anti-tumor effect, as they might also act as a cytokine sink?

Author response

As CD8⁺ T cells compete with Pmel-1 cells for cytokine availability, the depletion of endogenous CD8⁺ T cells will enhance the anti-tumor efficacy. This is supported by our unpublished data, in which the injection of polyclonal CD8⁺ T cells to Pmel-1-transferred mice reduced the therapeutic efficacy (see attached figure below). Despite the lack of Treg in the CD8⁺ T cell population, the competition for cytokines would negatively affect Pmel-1 activity. However, we anticipate that anti-tumor effect might increase if mice are injected with tumor-specific CD8⁺ T cells.

Anti-tumor effect of antigen-specific T cell and polyclonal CD4⁺ or CD8⁺ T cells. *Rag1* KO mice inoculated with 2×10^5 B16-F10 melanoma were infused with 2×10^6 cells of activated Thy1.1⁺ Pmel-1 CD8⁺ T (Pmel-1) cells. Some mice were additionally infused with 2×10^6 cells of polyclonal naive CD4⁺ or CD8⁺ T cells. Two days prior to cell transfer, mice were pre-conditioned with 4 Gy total body irradiation. Mice were daily treated with 10,000 IU IL-2 from day 5 to day 9.

Reviewer comment

- Are the same findings applicable to any other tumor model e.g. MC38 that is known to be inherently more immunogenic?

Author response

Unfortunately, we have only tried the current regimen in the B16-F10 model. We used Pmel-1 cells, which express well-defined melanoma-specific TCR, to evaluate the designed regimen in ACT efficacy. Nevertheless, we believe that this regimen can also be applied to other tumor models if CTX^{pre}/CD4^{post} is applied along with the transfer of tumor-specific CD8⁺ T cells. We have revised the manuscript to include this limitation of the current study and the anticipated results in other models (page 14, line 4 to 8).

Reviewer comment (detailed minor critique)

CD4post enhances ACT efficacy

Fig.1 Anti-CD4 post-conditioning increases anti-melanoma responses in adoptive T cell therapy - Ex-vivo priming of CD8+ T cells for only 2 days? This is not the standard protocol for pmel-1 experiments for many other researchers.

Author response: We have set this priming schedule to maximize the yield of activated Pmel-1 cells. Unlike other studies, we sorted the CD8⁺ fraction prior to activation. In addition, 5% splenocytes were added along with peptide stimulation to increase the viability of the sorted CD8⁺ T cells. The following figure indicates the unpublished data that compared three different activation conditions in terms of the changes in the viability and the expression of activation markers (4-1BB and CD25). The activation of whole splenocytes (left), sorted CD8⁺ T cells (center), and the mixture of CD8⁺ T cells + 5% splenocytes (right) were compared. Overall, while the marker expression increased until three days post-activation, the viability decreased sharply from day 3. The addition of 5% splenocytes in CD8⁺ T cells prevented the decrease in cell viability. Thus, we have used CD8⁺ T cells that were stimulated for 2 days under this condition.

Activation of Pmel-1 cells in various conditions. Pmel-1 cells were stimulated with gp100₂₅₋₃₃ peptide under three different conditions on day 0. Cells from lymph nodes and spleen of Thy1.1⁺ Pmel-1 mice were cultured without sorting (left), CD8-positive sorting (center), and CD8-positive

sorting in the presence of 5% splenocytes (right). Flow cytometry data indicates Thy1.1⁺ Pmel-1 cells that were stained with anti-4-1BB:PE and anti-CD25:FITC on the indicated time-point. Bar chart shows the viabilities of activated cells after stimulation with each condition on the indicated time-point.

- Please provide numbers of CD8⁺ T cells transferred.

Author response: We have revised all figure legends to provide the information regarding the number of transferred cells.

- Fig. 1A: no error bars in the spleen graph, and lymph node 300mg/kg condition, after 10 days. Does it mean only 1 mouse was analyzed?

Author response: We have conducted additional experiments with groups of 3 mice. The main figure has been replaced with the chart showing this result (Fig. 1a).

- Fig. 1C: shows very slow tumor growth (around day 15 in CTRL, day 20 for other conditions), maybe due to the low tumor initial number of B16 cells injected. This looks more like a protection than therapy experiments. Please provide a rationale for the quantity of tumor cells injected.

Author response: For consistency in experiments throughout this study, we used B16F10 cells with more than 80% viability and 90% confluence. The slow onset of tumor-growth is possibly due to the presence of CTX treatment, as the whole animal experiment included CTX treatment in this study. In addition, we set day zero as the point of tumor inoculation. The following figure from our unpublished data show the difference in tumor-growth and survival rate between the no treatment control and CTX treatment group. We revised the manuscript to include these information (page 17, line 14 to 19).

Survival and tumor growth rate under various conditions. C57BL/6 mice inoculated with 2×10^5 B16-F10 melanoma were injected with 2×10^6 ex-vivo primed Thy1.1⁺ Pmel-1 CD8⁺ T (ex-T) cells and IL-2 (10,000 IU). Two days before ex-T cell transfer, mice were pre-conditioned with 300 mg/kg CTX. The anti-CD4 group received weekly treatment of anti-CD4 antibody from day 10. The no Tx

group shows survival and tumor growth curve of mice that received only B16-F10 inoculation. The CTX group received only CTX conditioning without ex-T transfer.

- Furthermore, no CTX-only condition is shown, to compare with CTX/CD4.

Author response: As pointed by the reviewer, the CTX-only treated group has been indicated in Fig. 1c, d.

- In group CTX/ex-T/CD4, the two tumor-free mice seem not to have had tumor at all. Was the tumor presence asserted in any way?

Author response: Even in the mice that showed long-term tumor suppression, we observed melanoma-intrinsic blackish tissue at the site of B16-F10 injection 25 days after inoculation. Additionally, some tumor-free mice showed tumor-growth after day 100. These imply the presence of tumors at the early time-point in the tumor-free mice.

- Please show tumor area (mm²) rather than volume (mm³): this is the more standard and less biased way of measuring the responses in pmel-1 model.

Author response: As we set the guideline of animal euthanasia based on tumor volume (mm³), we have decided to use volume instead of area (mm²) in the current study. Please note that we have confirmed that the conversion from volume to area did not change the statistical significance in our major *in vivo* data nor the conclusion of this study.

Fig. 2. Anti-CD4 post-conditioning accelerates the division and intratumoral infiltration of ex vivo primed and endogenous CD8⁺ T cells.

- Please comment on selecting day 25 as endpoint.

Author response: Longitudinal analysis of the peripheral blood of CTX^{pre}/CD4^{post}-experienced mice showed the significant change in the proportion of ex-T/en-T cells on day 24 (see attached figure below). To investigate characteristic changes of CD8⁺ T cells at the earliest time point, we have analyzed lymphoid tissues on day 25.

Change in ratio of ex-T and en-T in peripheral blood. C57BL/6 mice were treated with CTX/ex-T or CTX/ex-T/CD4 regimen (as in Fig. 1b) and used for this analysis. The proportion of transferred Thy1.1⁺ Pmel-1 CD8⁺ T (ex-T) and endogenous Thy1.1⁻ CD8⁺ T cells (en-T) in peripheral blood was analyzed using flow cytometry at the indicated time point.

- Please provide FACS gates for CD44^{hi}CD62L^{lo} cells (in supp. Fig. this specific population is not shown).

Author response: We have added the gating strategy in Supplementary Fig. 2c.

- Correct legend in Supp.Fig.2B, to be consistent with legend in Fig.2C

Author response: We have corrected the legend (which was also moved to Supplementary Fig. 4b).

- Please comment on the higher tumor cell dose used in Fig.2G-J

Author response: We injected larger number of tumor cells to establish analyzable tumor size in both CTX^{pre} and CTX^{pre}/CD4^{post} mice. We have revised all figure legends to show the transferred cell number.

- Fig.2G: Please provide numbers of CD8⁺ T cells injected.

Author response: We have revised all figure legends to show the transferred cell number.

- Fig.2J: please change colors, or line shape, of histograms, to make it more distinguishable.

Author response: We apologize for the difficulty. We have revised the shading and line shape to make the figure clearer (Supplementary Fig. 2g).

Fig. 3. Anti-CD4 post-conditioning increases the differentiation and tumor-reactivity of endogenous CD8+ T cells.

- Please provide FACS gates (dot plots) for populations analyzed: 4-1BB, CD69, CD44hiCD62Llo

Author response: We have added the gating strategy in Supplementary Fig. 2a-c.

- Supp.Fig.3A/C: FACS gate for IL-2 is not convincing – there is very little signal. Please provide the negative control used to set it up (as well as for other cytokines). Please also provide gate strategies for Supp.Fig.3B/D.

Author response: We have added the negative control used in the gating strategy in Supplementary Fig. 5a, c, e.

- Fig.3E: please choose a different color scheme that could be easier to read for all readers (the reviewer is colorblind).

Author response: We apologize for the inconvenience. We have changed the color scheme for colorblind-friendly visualization (Fig. 3b).

- Fig.3D: same comment as Fig.2J

Author response: We have revised the shading and line shape to make the figure clearer (Supplementary Fig. 2d).

- Fig.3H: the message here is difficult to get at first glance, maybe the authors might want to use a different type of graph.

Author response: We have changed the type of graph for TCR repertoire. Shannon evenness was presented in scatter plots (Fig. 3e).

Fig. 4. Anti-CD4 post-conditioning enriches IL-18R α ^{hi} endogenous CD8⁺ T cells with potent effector potential.

- Please provide numbers of CD8⁺ T cells/En-T cells injected.

Author response: We have revised all figure legends to show the transferred cell number.

- Fig.4B: did the 2 tumor-free mice actually show any growth of tumor cells?

Author response: As mentioned in an earlier response, we observed melanoma-intrinsic blackish tissue at the site of B16-F10 injection in these mice 25 days after tumor inoculation.

- Fig.4G: please correct legend, are IL-18R α ⁻ and IL-18R α ⁺ respectively IL-18R α ^{mid} and IL-18R α ^{high}?

Author response: The cells were categorized into three subsets (lo, mid, and hi) and subjected to the polyfunctionality evaluation (Fig. 5a).

- Fig.4G: See comments Fig.3E and Fig.3A/C.

Author response: We have added the negative control used in the gating strategy (Supplementary Fig. 5a, c, e) and changed the color scheme for colorblind-friendly visualization (Fig. 5a).

Fig. 5. IL-18R α ^{hi} endogenous CD8⁺ T cell subset with innate-like property has potent anti-tumor effect.

- Please provide numbers of CD8⁺ T cells injected.

Author response: We have revised all figure legends to show the transferred cell number.

- In the flow through, 20% of the cells are IL-18R α ^{high}. Was this contamination addressed while analyzing TCR and gene expression?

Author response: In the original figure legend, we labeled the subset as “flow through” because the magnetically sorted population included 20% of IL-18R α ^{hi} subset. To avoid confusion, we have revised the manuscript to indicate the sorted populations as “enriched for high IL-18R α -expression” vs. “other cells in the flow through” (page 7, line 20 to 21).

- Where iNKT and MAIT characterized by their phenotype, in addition to TCR clonotypes and gene enrichment?

Author response: We appreciate the reviewer's suggestion. By using the CD1d/ α GalCer tetramer, we confirmed that iNKTs express IL-18R α under the influence of CTX^{pre}/CD4^{post} (see attached figure below). Nevertheless, throughout the additional studies performed during the revision, we have observed that tumor-specific IL-18R α ^{hi} en-T/ex-T cells were the key components that elicited robust anti-tumor response, while IL-18R α ^{hi} iNKT/MAIT was the bystander product resulting from the CTX^{pre}/CD4^{post}-induced milieu. We have added the detailed discussion in the revised manuscript (page 13, line 18 to page 14, line 1).

Binding of CD1d/ α GalCer tetramer to En-T cells. En-T cells from CTX^{pre}- and CTX^{pre}/CD4^{post}-experienced mice were stained with CD1d/ α GalCer tetramer:PE and anti-IL-18R α :APC. (Left) Representative staining data. (Right) Summary of total staining data (n=5 mice per group). **, p<0.01; ***, p<0.001; two-tailed unpaired Student's t-test.

Discussion:

The activated en-T cells that express high level of IL-18R include iNKT, MAIT, NKG2D+ T cells, and TVM, which show cytokine-mediated anti-tumor responses.

- There is no cytokine-mediated anti-tumor response, related to these cells, shown.

Author response: We have found that IL-18R α ^{hi} en-T cells elicited anti-tumor response via TCR-MHC I interaction (Fig. 5c). Furthermore, IL-18R α ^{hi} ex-T cells, which have tumor-specific TCR, drove tumor-suppressive effect in an IL-18-dependent manner (Fig. 5h, j, k). We anticipated that the iNKTs and MAITs population expanded in response to the bystander proliferation caused by the CTX^{pre}/CD4^{post}-induced lymphopenic milieu. This is supported by previous reports, in which iNKTs and MAITs expressed high level of receptor for IL-7 (PMID: 18354162 and 23447689). It is noteworthy that IL-7-FC profoundly increased the IL-18R α ^{hi} ex-T/en-T subset in the current study (Fig. 6h). Similarly, NKG2D and T_VM-like phenotypes were caused by the same lymphopenic condition, as the related markers are frequently upregulated by TCR-independent cytokine-dependent activation (PMID: 23523350, 25727290, 29305140, and 30413351). We have revised the discussion

to include this point (page 13, line 18 to page 14, line 4).

General Comments:

- *Were other models tested? And why, with the pmel mouse model, no vaccination was used?*

Author response: To test the designed regimen in the ACT model, we have used Pmel-1 cells, which are highly melanoma-specific TCR-expressing T cells, and B16-F10 without vaccination. We have added this limitation in the Discussion (page 14, line 4 to 8).

Methods:

Cells

- *Please state viability and confluence of B16 cells at the moment of injection.*

Using CD8 microbeads (positive selection beads; Miltenyi Biotech, CA, USA), CD8+ T cells were isolated from lymphocytes. For ex vivo priming, Thy1.1+ Pmel-1 CD8+ T cells were further stimulated with 5 µg/mL of human gp10025–33 peptide (KVPRNQDWL; Peptron, Daejeon, Korea) in the presence of 5% splenocytes for 2 days.

Author response: For consistency in experiments throughout this study, we used B16F10 cells with more than 80% viability and 90% confluence. We have revised the manuscript to include this information in the Methods (page 17, line 14 to 19).

- *Were the CD8+ T cells isolated only from lymphocytes or, as stated in the line before, also from spleen? Is the 2 days enough to get a proper priming of CD8+ T cells? Were longer conditions tested? And which medium was used for the priming? Finally, were the splenocytes irradiated?*

- *Why such a short stimulation?*

Author response: We isolated the CD8⁺ fraction from the cells obtained from the lymph nodes and spleen. As shown in the previous response to the reviewer's comment, we have tested priming length between days 0 to 5. The priming of 2 days was decided based on the optimal balance between the activation and viability of CD8⁺ T cells. We have used 10% FBS-supplemented RPMI media and 5% splenocytes (non-irradiated). We have revised the manuscript to include this information in the Methods (page 16, line 8 to 14).

Adoptive transfer models

On day 5, mice received intravenous injections of CD8+ T cells.

- *Please provide number of injected cells, and localization of the injection.*

Author response: The number of CD8⁺ T cells injected has been indicated in each figure legend. We

injected these cells via the tail vein throughout this study. We have revised the manuscript to include this information in the Methods (page 17, line 21).

- Please add a paragraph related to Fig.4A

Author response: We have revised the manuscript to include the detail of this information in the Methods (Fig. 5b, g in the revised manuscript; page 18, line 5 to 16).

Flow cytometry

The isolated cells were placed in the well (5×10^5 /well) of a 96-well round-bottom plate, treated with a Cell Stimulation Cocktail and Protein Transport Inhibitor Cocktail (Thermo Fisher Scientific, MA, USA), and then incubated for 16 h.

- This is not a typical length of in vitro stimulation with this cocktail (see package insert). Overall PMI/ionomycin and protein transport inhibitors are toxic to T cells if left in culture for too much time, likely biasing the results.

Author response: We appreciate the reviewer's suggestion. We have performed additional experiments using 6 h as the duration of the incubation. All intracellular cytokine staining data in the revised manuscript show the results after 6 h of stimulation.

Reviewers' Comments:

Reviewer #1:

Remarks to the Author:

The authors have successfully answered my previous comments and those of the other reviewers.

Reviewer #2:

Remarks to the Author:

I congratulate the authors for their thorough and considered reply to reviewer comments. They have addressed several of the major points raised in the first round of review including the dependency for MHC-I in the effect mediated by IL-18Rhi cells.

I believe the remaining issues should be addressed:

- Whilst the authors provide new data with aGalCer staining that confirms the enhanced presence of NKT cells, similar data are not provided with the MAIT cell tetramer.

The TRAV1/TRAJ33 sequence used to define MAIT cells here is known to be shared by other non-MR1 restricted T cells, that by definition are not MAIT cells. Therefore I believe this needs to be explicitly stated as a major caveat of this interpretation or this data removed. I would recommend the later as these cells are not interrogated further in this study.

-In a related point the authors refer to this TCR (and an NKT TCR) in Supp Figure 6 that is expanded following CD4 depletion but this is not placed in the context of overall TCR diversity i.e. Are these the most expanded clones? I think this comparison is important if the authors choose to include the data.

- The authors have addressed the importance of the IL-18Rhi cells through some intricate transfer experiments. In Figure 5C the authors conclude that "Initial tumour growth rate in IL-18Rhi-transferred mice was significantly lower than that of mice treated with CTX/CD4post-exposed en-T cells. However, I am not sure this is supported by the data as the difference does not look to be (and is not denoted as) statistically different. I would therefore recommend softening this statement.

Having reviewed the authors' responses to reviewer 3, I believe the following point should be clarified:

The reviewer requests further information on the subset of CD4+ T cells involved in the suppression of the IL-18Ralphahi phenotype. The authors refer to an experiment where Treg depletion is shown to have no effect on the emergence of this subset. They conclude that this result "suggests the importance of Treg in the formation of this milieu".

They also state that "Interestingly, the depletion of Tregs instead of whole CD4+ cells, also increased IL-18Rahi ex-T cell frequency (Fig. 6a)." This does not seem to be borne out by the data where no significant difference is shown in this figure.

Thus, an alternative explanation is that total CD4+ depletion simply results in enhanced lymphopenia over and above specific depletion of Tregs and thus the effect is not related to Tregs suppressive activity per se. I believe this should be discussed and clarified.

The other major requests by reviewer 3 pertain to additional experiments such as the addition of extra tumor models. Whilst these would be interesting and further support the conclusions, they would result in substantially more experiments to be performed.

All other minor comments have been addressed adequately with the possible exception regarding the request to change the color scheme for figure 3b. I am not well-placed to comment on this as I am not color blind myself, but I think that the choice of green and red could be problematic. I refer

to the journal policy on this issue.

We thank the reviewer for comments for improving our manuscript. Based on the suggestions we have revised the manuscript for clarity. Please find the point-by-point responses to the reviewer's comments as follows.

Reviewer #2 (Remarks to the Author):

I congratulate the authors for their thorough and considered reply to reviewer comments. They have addressed several of the major points raised in the first round of review including the dependency for MHC-I in the effect mediated by IL-18Rhi cells.

I believe the remaining issues should be addressed:

- Whilst the authors provide new data with aGalCer staining that confirms the enhanced presence of NKT cells, similar data are not provided with the MAIT cell tetramer.

The TRAV1/TRAJ33 sequence used to define MAIT cells here is known to be shared by other non-MRI restricted T cells, that by definition are not MAIT cells. Therefore I believe this needs to be explicitly stated as a major caveat of this interpretation or this data removed. I would recommend the later as these cells are not interrogated further in this study.

-In a related point the authors refer to this TCR (and an NKT TCR) in Supp Figure 6 that is expanded following CD4 depletion but this is not placed in the context of overall TCR diversity i.e. Are these the most expanded clones? I think this comparison is important if the authors choose to include the data.

Author response

The importance of innate-like CD8⁺ T cells (i.e., NKT and MAIT) has been decreased since the last revision, because IL-18R^{hi} Pmel-1 cells were found to be crucial for the enhanced anti-tumor effect. Therefore, as the reviewer suggested, we have removed the data and revised the manuscript regarding the interpretations on T cells with NKT/MAIT TCRs (page 7, line 23 to 27 and page 13, line 26 to page 14, line 1).

- The authors have addressed the importance of the IL-18Rhi cells through some intricate transfer experiments. In Figure 5C the authors conclude that "Initial tumour growth rate in IL-18Rhi-transferred mice was significantly lower than that of mice treated with CTX/CD4post-exposed en-T cells. However, I am not sure this is supported by the data as the difference does not look to be (and is not denoted as) statistically different. I would therefore recommend softening this statement.

Author response

As per the reviewer's recommendation, we have deemphasized the anti-tumor effect of IL-18R^{hi} en-T cells (page 8, line 26 to page 9, line 1).

Having reviewed the authors' responses to reviewer 3, I believe the following point should be clarified:

The reviewer requests further information on the subset of CD4⁺ T cells involved in the suppression of the IL-18R α ^{hi} phenotype. The authors refer to an experiment where Treg depletion is shown to have no effect on the emergence of this subset. They conclude that this result "suggests the importance of Treg in the formation of this milieu".

They also state that "Interestingly, the depletion of Tregs instead of whole CD4⁺ cells, also increased IL-18R α ^{hi} ex-T cell frequency (Fig. 6a)." This does not seem to be borne out by the data where no significant difference is shown in this figure.

Thus, an alternative explanation is that total CD4⁺ depletion simply results in enhanced lymphopenia over and above specific depletion of Tregs and thus the effect is not related to Tregs suppressive activity per se. I believe this should be discussed and clarified.

Author response

Our intention with the following description "Notably, Treg depletion, instead of whole CD4⁺ cell depletion, did not result in any significant difference in IL-18R α ^{hi} frequency compared to that of the control group (full factors)" was that the proportion of IL18R α ^{hi} T cells in the Treg-depleted group was not significantly different from that in the IL18R α ^{hi}-enriched CTX^{pre}/CD4^{post} group, which means Treg depletion had a similar effect as the CTX^{pre}/CD4^{post} treatment. We apologize for the confusion. To avoid further confusion, we have revised the manuscript (page 10, line 23 to 24).

The other major requests by reviewer 3 pertain to additional experiments such as the addition of extra tumor models. Whilst these would be interesting and further support the conclusions, they would result in substantially more experiments to be performed.

Author response

Thank you for your understanding regarding this issue. As the reviewer pointed out, one of the limitations of this study lies in B16F10 melanoma-restricted validation of the CTX^{pre}/CD4^{post} regimen. We hope to find the effect of the regimen in other tumor models in the future as described in Discussion (page 14, line 6 to 8).

All other minor comments have been addressed adequately with the possible exception regarding the request to change the color scheme for figure 3b. I am not well-placed to comment on this as I am not color blind myself, but I think that the choice of green and red could be problematic. I refer to the

journal policy on this issue.

Author response

We appreciate the comment. We chose the color scheme according to the suggestion by Wong et al. (Points of view: Color blindness. [2011] *Nature Methods* **8**:441). The expected images for three different color-blindness were simulated using the following simulator: <https://www.color-blindness.com/coblis-color-blindness-simulator/>. We have attached the sample images for Figure 3b as follows: